# An open label randomized controlled trial of tamoxifen combined with amphotericin B and fluconazole for cryptococcal meningitis

Nguyen Thi Thuy Ngan[1,2], Nhat Thanh Hoang Le[2], Nguyen Ngo Vi Vi[2], Ninh Thi Thanh Van[2], Nguyen Thi Hoang Mai[2], Duong Van Anh[2], Phan Hai Trieu[2], Nguyen Phu Huong Lan[3], Nguyen Hoan Phu[2], Nguyen Van Vinh Chau[3], David G Lalloo[4], William Hope[5], Justin Beardsley[6,7], Nicholas J White[8,9], Ronald Geskus[2,9], Guy E Thwaites[2,9], Damian Krysan[10], Luong Thi Hue Tai[3], Evelyne Kestelyn[2,9], Tran Quang Binh[1], Le Quoc Hung[1], Nguyen Le Nhu Tung[3], Jeremy N Day[2,9]*

[1]Department of Tropical Medicine, Cho Ray Hospital, Ho Chi Minh City, Viet Nam; [2]Oxford University Clinical Research Unit, Ho Chi Minh City, Viet Nam; [3]The Hospital for Tropical Diseases, Ho Chi Minh City, Viet Nam; [4]Liverpool School of Tropical Medicine, Liverpool, United Kingdom; [5]Centre of Excellence in Infectious Disease Research, Institute of Translational Medicine, Liverpool University, Liverpool, United Kingdom; [6]The University of Sydney, Marie Bashir Institute, NSW, Camperdown, Australia; [7]Westmead Institute for Medical Research, Westmead, Australia; [8]Mahidol Oxford Research Unit, Faculty of Tropical Medicine, Mahidol University, Bangkok, Thailand; [9]Centre for Tropical Medicine and Global Health, Nuffield Department of Medicine, University of Oxford, Oxford, United Kingdom; [10]Department of Paediatrics and Microbiology/Immunology, Carver College of Medicine, University of Iowa, Iowa City, United States

*For correspondence: jday@oucru.org

## Abstract

**Background:** Cryptococcal meningitis has high mortality. Flucytosine is a key treatment but is expensive and rarely available. The anticancer agent tamoxifen has synergistic anti-cryptococcal activity with amphotericin in vitro. It is off-patent, cheap, and widely available. We performed a trial to determine its therapeutic potential.

**Methods:** Open label randomized controlled trial. Participants received standard care – amphotericin combined with fluconazole for the first 2 weeks – or standard care plus tamoxifen 300 mg/day. The primary end point was Early Fungicidal Activity (EFA) – the rate of yeast clearance from cerebrospinal fluid (CSF). Trial registration https://clinicaltrials.gov/ct2/show/NCT03112031.

**Results:** Fifty patients were enrolled (median age 34 years, 35 male). Tamoxifen had no effect on EFA ($-0.48$log10 colony-forming units/mL/CSF control arm versus $-0.49$ tamoxifen arm, difference $-0.005$log10CFU/ml/day, 95% CI: $-0.16$, 0.15, p=0.95). Tamoxifen caused QTc prolongation.

**Conclusions:** High-dose tamoxifen does not increase the clearance rate of *Cryptococcus* from CSF. Novel, affordable therapies are needed.

**Funding:** The trial was funded through the Wellcome Trust Asia Programme Vietnam Core Grant 106680 and a Wellcome Trust Intermediate Fellowship to JND grant number WT097147MA.

## Introduction

Cryptococcal meningitis is a leading cause of death in HIV-infected patients, with an estimated 223,000 cases in 2014 (*Rajasingham et al., 2017*). The vast majority of infections are due to *C. neoformans*, and occur in low-income tropical settings. Current international guidelines recommend initial induction treatment with amphotericin combined with flucytosine, followed by consolidation therapy with fluconazole (*World Health Organization, 2018*). This combination delivers the fastest rates of clearance of yeast from cerebrospinal fluid (CSF) and the best survival rates (*Day et al., 2013*; *Molloy et al., 2018*). However, even on this gold standard therapy, 30% of patients will die within 10 weeks of diagnosis (*Day et al., 2013*; *Molloy et al., 2018*). Adjunctive therapy with corticosteroids, which has proven beneficial in other forms of meningitis, results in worse outcomes (*Beardsley et al., 2016*).

Cryptococcal meningitis can also occur in HIV-uninfected patients, including immunocompetent people and those with other causes of immunosuppression. Survival rates are similar to those seen in HIV-infected patients. There are few data from randomized controlled trials to guide treatment in these circumstances. In Vietnam around 20% of cases of cryptococcal meningitis are in HIV-uninfected patients (*Chau et al., 2010*). Disease is predominantly due to the *C. neoformans* VNIa-5 lineage; *C. gattii* is responsible for around 25% of cases (*Chau et al., 2010*; *Ashton et al., 2019*; *Day et al., 2011*; *Day et al., 2017*).

There has been little progress in development of antifungal drugs for cryptococcal meningitis. Amphotericin and flucytosine are each more than 60 years old; the last novel drug class developed was the azoles, introduced 30 years ago. Access to flucytosine is severely restricted by availability and cost, meaning it is rarely used where disease burden is highest. Despite being off-patent, it has been subject to extraordinary price rises in recent years, with a 2-week course now costs around 30,000 USD in the USA (*Merry and Boulware, 2016*). Flucytosine is an unattractive prospect for generic manufacturers, because the location of the majority of patients and the few indications outside cryptococcal disease promise only limited financial returns. These same factors hamper the development of novel treatments for cryptococcal disease, and have driven interest in drug re-purposing (*Butts et al., 2014*; *Dolan et al., 2009*; *Zhai et al., 2012*). Re-purposing can be a solution for neglected diseases provided the new indication accounts for only a minority of total prescriptions, and the de facto indications are sufficiently prevalent to ensure availability, price stability, and affordability.

Tamoxifen, a selective estrogen receptor modulator used to treat breast cancer, has anti-cryptococcal activity, appearing to act synergistically when combined with other antifungals against the type strain in vitro, and to be fungicidal when combined with fluconazole in the mouse infection model (*Butts et al., 2014*; *Dolan et al., 2009*). We found it to act synergistically with amphotericin against two-thirds of clinical isolates of *Cryptococcus neoformans* and *C. gattii* from our archive and to have an additive interaction when combined with fluconazole in vitro (*Hai et al., 2019*).

Tamoxifen is concentrated in brain tissue (10- to 100-fold compared with plasma) and macrophage phagosomes (a site of growth for *Cryptococcus* spp.), is off-patent, cheap (~10US cents/tablet) and widely available (*Lien et al., 1991a*; *Lien et al., 1991b*). Therefore, it is a promising candidate for the treatment of cryptococcal meningitis. Pharmacokinetic data suggest that doses 5- to 10-fold that used in breast cancer (typically 30 mg/day) should deliver plasma concentrations of tamoxifen greater than the Minimum Inhibitory Concentration 90 (MIC90 16 µg/mL) of Vietnamese clinical isolates (*Lien et al., 1991a*). Such doses have been used, and well-tolerated, in small cell lung cancer, desmoid tumours, and prostate cancer. These illnesses have comparable or better 1-year survival rates than cryptococcal meningitis (*Ngan et al., 2019*). While generally well-tolerated, acute side effects that could be detrimental from short-course treatment include QT prolongation of the cardiac de/repolarization cycle, although the risk of life-threatening arrhythmias appears to be low (*Grouthier et al., 2018*).

In Vietnam induction treatment for cryptococcal meningitis consists of amphotericin combined with fluconazole, consistent with WHO recommendations where flucytosine is unavailable (*World Health Organization, 2018*). However, this combination is less effective than amphotericin with flucytosine, resulting in slower rates of fungal clearance and worse survival rates (*Day et al., 2013*; *Molloy et al., 2018*). The relationship between the rate of fungal clearance from CSF and survival is generally robust; improving the potency of antifungal therapy is likely to be an effective way

to reduce deaths (*Day et al., 2013*; *Molloy et al., 2018*; *Beardsley et al., 2016*). The rate of clearance of yeast from CSF associated with an antifungal treatment (the early fungicidal activity, EFA) is a sensitive measure able to detect differences between treatment regimens likely to be associated with survival benefits with far fewer patients than studies powered to survival itself (*Brouwer et al., 2004*). Small studies powered to this endpoint can serve to filter treatment regimens that can be taken forward in larger trials (*Brouwer et al., 2004*; *Bicanic et al., 2009*). We performed an open-label randomised controlled trial to determine whether combining tamoxifen with amphotericin B and fluconazole results in enhanced EFA in HIV infected and uninfected patients with cryptococcal meningitis, and to generate safety data as a prelude to a larger trial powered to mortality (*Ngan et al., 2019*).

## Materials and methods

### Study design and participants

The study design is described in detail in the published protocol (*Ngan et al., 2019*). In brief, we enrolled 50 patients in two hospitals in Ho Chi Minh City – the Hospital for Tropical Diseases and Cho Ray Hospital. Eligible adult patients ($\geq$18 years of age) had a clinical syndrome consistent with cryptococcal meningitis and one or more of: (1) positive cerebrospinal fluid (CSF) India ink; (2) *C. neoformans* cultured from CSF or blood; (3) positive cryptococcal antigen Lateral Flow Antigen Test (LFA) in CSF. All patients were tested for HIV infection in accordance with standard of care. We excluded patients who were pregnant, had a history of thromboembolic disease, had received more than 4 days of anti-cryptococcal antifungal therapy, had any other indication for tamoxifen, had renal failure, or a rate-corrected (Framingham formula) QT interval >500ms. Written informed consent was obtained from all patients or their representatives.

### Interventions

Patients were randomized to receive either standard of care induction antifungal therapy or standard of care plus tamoxifen. Standard of care antifungal therapy consisted of intravenous amphotericin B deoxycholate 1 mg/kg/day (Amphotret, Bharat Serums and Vaccines, India) combined with oral fluconazole 800 mg/day (Zolmed, Glomed Pharmaceuticals, Vietnam) for the first 14 days following randomization. Tamoxifen (Nolvadex, AstraZeneca UK Ltd) 300 mg/day was given orally. Amphotericin was infused over 4 hr after prehydration with normal saline and potassium supplementation (*Khoo et al., 1994*). Fluconazole and tamoxifen were administered simultaneously. All medication was directly observed while the patient was in hospital; all participants were in-patients for at least the first 14 days of the study.

Following induction therapy all patients received fluconazole 800 mg once daily for 8 weeks. HIV-infected patients received daily pneumocystis prophylaxis with trimethoprim– sulfamethoxazole. Antiretroviral therapy was instituted 5–6 weeks after diagnosis via the national treatment programme.

### Randomization

Randomization was in a ratio of 1:1, in blocks of 4 or 6, stratified by HIV serostatus (rapid test) and treating centre. The computer generated randomization list was password protected and stored on a secure server to which only the study pharmacist had access. Enrolment logs specific to each centre were used to assign patients to the next available sequential number and corresponding sealed treatment pack.

### Outcome measures

The primary outcome was Early Fungicidal Activity (EFA), defined as the rate of decline in culturable yeast from CSF over the first 2 weeks following randomization.

Secondary outcomes included survival until 10 weeks after randomization, disability at 10 weeks, frequency of grade 3, 4 or serious adverse events, immune reconstitution inflammatory syndrome (IRIS), QTc prolongation, visual deficit at 10 weeks, and time to new neurological events. Adverse events were defined according to the Common Terminology Criteria for Adverse Events (CTCAE) and categorized according to the Medical Dictionary for Regulary Activities system organ class. We

categorized prolonged QTc intervals using this classification as normal (<450 ms for males, <460 ms for females), mildly prolonged (grade 1 or 2, $\geq$450 ms for males or $\geq$460 for females but $\leq$500 ms) and grade 3 or 4 (>500ms). Disability at 10 weeks was categorised as good, intermediate, poor, or death, as described previously (*Day et al., 2013*; *Beardsley et al., 2016*).

## Monitoring and laboratory investigations

Lumbar puncture was performed on study entry, days 3, 7, and 14 following randomization, and more frequently if indicated. Fungal burden was determined as previously described (*Day et al., 2013*). Twelve-lead electrocardiograms were recorded twice daily (10 s at 50 mm/sec), immediately before and 2 hr after administration of tamoxifen during the first 14 days, and on days 21 and 28. The QT interval was manually determined by measuring the interval in three limb and three chest leads, to calculate the median. The median QT interval was corrected (QTc) for rate using the Framingham formula[20]. Calmodulin inhibitors such as tamoxifen have previously been suggested to inhibit CD4 cell apoptosis in HIV-infected patients (*Pan et al., 1998*). CD4 counts were measured at baseline and at study week 10. The full laboratory investigation schedule is detailed in the published protocol (*Ngan et al., 2019*). Outpatient assessments with medication review were performed weekly until 4 weeks and at the completion of 6 and 10 weeks; more frequent review occurred if clinically indicated. Adherence following hospital discharge was assessed using pill counts. *Cryptococcus* isolates were typed using URA5-RFLP and underwent (microbroth) antifungal susceptibility testing as per CLSI guidelines (*CLSI, 2002*; *Franzot et al., 1997*). Previously tested clinical isolates were included as controls.

## Sample size

Sample size considerations were based on two separate simulation experiments using data from our previously published trials in cryptococcal meningitis (*Day et al., 2013*; *Beardsley et al., 2016*). The estimated power was based upon 10,000 repetitions of each experiment. The full methodology is available within the published protocol (*Ngan et al., 2019*). Based on these simulations, enrolling 25 subjects per treatment group provided 80% and 90% power to detect a difference in EFA of −0.11 or −0.13 log10 colony-forming units/ml/day, respectively. This size of effect has previously been associated with survival benefit (*Day et al., 2013*; *Beardsley et al., 2016*).

## Statistical analysis

For the primary outcome, all recorded longitudinal quantitative fungal count measurements up to day 17 following randomization (allowing for some delays in the day 14 sampling) were included in the analysis. EFA, defined as the decline in fungal count (slope), was modeled based on a joint model consisting of a survival model and a linear mixed effects model with longitudinal log10 CSF quantitative culture fungal counts as the outcome. In the linear mixed effect model, we modeled the treatment groups and the time since enrolment and their interaction as fixed covariates. We used random patient-specific intercepts and slopes. The model was implemented in a Bayesian framework using Rstan. It allows appropriate handling of detection limits with longitudinal measurements and also allows adjustment for informative dropout due to early death within the first 17 days following randomization (*Stan Development Team, 2019*; *R Development Core Team, 2018*).

For the secondary outcomes, overall survival was visualized using Kaplan-Meier curves for each treatment arm and the comparison between them was based on the Kaplan Meier estimates of 10-week mortality. The percentage of individuals with disabilities at 10 weeks and with adverse events of grade 3 or 4 were compared using the chi-squared test; if the expected value of any cell was less than one then Fisher's exact test was used (*Campbell, 2007*). We presented the median (IQR) of the difference in CD4 counts over 10 weeks and compared their distributions using the Mann-Whitney-Wilcoxon rank sum test. We compared the trend in QTc over the period of study drug administration (i.e. the first 14 days) between the two treatment arms using a linear mixed effect model which allowed for different non-linear trends between the pre-dose and post-dose measurements. We then used the output of the fitted linear mixed effect model to compute the differences in QTc between treatment arms by study day, separately for pre-dose and 2 hr post-dose measurements. Further details of the analytical approach are available in the the Supplementary Appendix in the Statistical Analysis Plan.

## Results

### Trial recruitment

The study recruited between October 2017 and May 2018. We screened 70 patients, enrolling 50 (40 HIV infected; 10 HIV uninfected) with 24 assigned to the intervention arm and 26 assigned to the

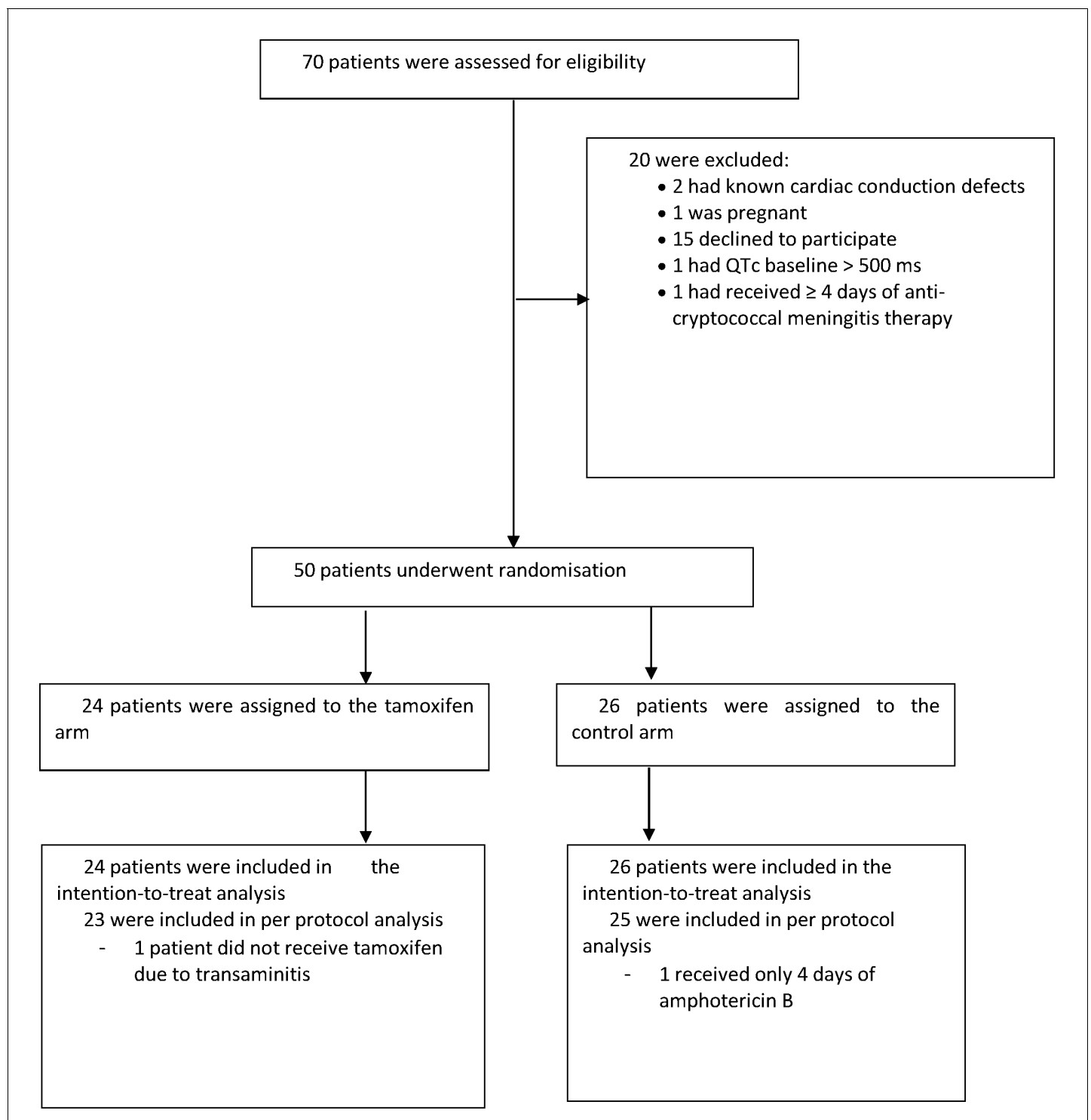

**Figure 1.** Trial flow chart: enrollment, randomization, and follow-up.

control arm. Reasons for exclusion are shown in the study flow diagram (see *Figure 1*). One patient who was assigned to the intervention arm did not receive tamoxifen because of severe transaminitis.

## Baseline characteristics

The baseline characteristics of the patients were broadly balanced between treatment groups. There were slightly more patients with normal Glasgow coma scores in the control group than in the intervention group (24 of 26 versus 19 of 24, see *Table 1*).

## Primary outcome

There was no detectable difference in the early fungicidal activity (EFA) of the two treatment regimens (see *Figure 2A*). In the intention-to-treat analysis, the rates of fungal decline per day were $-0.48$ and $-0.49$ $\log_{10}$colony-forming units (CFU)/ml/day in the control and tamoxifen groups respectively (difference $-0.005$ $\log_{10}$ CFU/ml/day, 95% CI: $-0.16$, 0.15); p-value = 0.95, (see *Table 2*). There was no detectable difference in EFA in the per-protocol population analysis, or by HIV infection status (see *Table 2*).

## Secondary endpoints

The secondary outcomes in terms of mortality, disabilities, and change in CD4 count are summarized in *Table 3*. Death occured in 8 of 24 patients in the tamoxifen group and 7 of 26 in the control group (Kaplan-Meier mortality estimates 34% and 27% respectively, risk difference 6.5%; 95% confidence interval [CI], $-19.2$ to 32.1%; P=0.62 *Figure 2B*). Fewer patients in the tamoxifen arm were classified as having a good outcome at 10 weeks compared with the control arm (9% versus 36%). We found no difference in change in CD4 counts in HIV patients by study arm over the 10-week period of follow-up (see *Table 3*).

The number of patients having grade 3 or 4 adverse events were similar between treatment arms (see *Table 4*), with the exception of QTc prolongation events. Eight patients had grade 3 or 4 QTc prolongation events in the tamoxifen arm, compared with one in the control arm (p=0.02). The trend and difference in QTc intervals over the first 2 weeks of treatment are shown in *Figure 3*. Tamoxifen resulted in QTc prolongation over the 2-week treatment period (p<0.001).

Three patients in the tamoxifen arm had grade 3 or 4 ventricular extra-systole events compared with none in the control arm (p=0.21). A 33-year-old male patient who had received tamoxifen suffered a cardiorespiratory arrest following a convulsion on day 21 of the study. He had no history of pre-existing cardiac disease. His ECG on admission had been normal with a QTc of 409 ms, and when performed routinely on the morning of day 21 showed mild sinus bradycardia (57 beats/minute) and a QTc interval of 477 ms. The arrest was not associated with ventricular arrhythmia although he had had grade 3 prolongation of QTc during the first 14 days of the study, which had resolved following tamoxifen interruption.

## Microbiology and susceptiblity testing

All HIV infected patients, and seven HIV uninfected patients, had meningitis due to *Cryptococcus neoformans* molecular group VNI. Three HIV uninfected patients had disease due to *Cryptococcus gattii* (VGI). All isolates underwent susceptibility testing. The MIC90 of amphotericin B and fluconazole were 2 mg/L and 4 mg/L, respectively. The MIC90 of tamoxifen was 8 mg/L. We estimated the presence of drug interactions by calculating the fractional inhibitory concentration index (FICI) for each isolate. This was ≤0.5 (suggestive of a possible synergistic interaction) for tamoxifen combined with amphotericin in six isolates (12%), and for tamoxifen combined with fluconazole in two isolates (4%).

## Discussion

We wanted to determine whether tamoxifen could be repurposed as an affordable treatment for cryptococcal meningitis. Our study was powered to detect an increase in the rate of yeast clearance of at least $-0.11$ log10 CFU/ml/day when tamoxifen was added to standard of care therapy. Differences of this order of magnitude are associated with improved survival in patients in low-income settings (*Day et al., 2013*; *Molloy et al., 2018*; *Beardsley et al., 2016*). Despite having previously

**Table 1.** Clinical and investigation characteristics of patients at study entry.

| Characteristic | Total | Tamoxifen | Total | Control |
|---|---|---|---|---|
| | N | N (%) or IQR* | N | N (%) or IQR* |
| Male sex | 24 | 17 (71) | 26 | 18 (69) |
| Median age in years | 24 | 35 (31, 39) | 26 | 32 (25, 35) |
| History of intravenous drug use | 24 | 3 (13) | 26 | 3/26 (12) |
| HIV infection | 24 | 19 (83) | 26 | 21/26 (81) |
| Current antiretroviral-therapy use | | | | |
| None | 24 | 18 (75) | 26 | 22 (84) |
| ≤3 months duration | 24 | 4 (17) | 26 | 2 (8) |
| >3 months duration | 24 | 2 (8) | 26 | 2 (8) |
| Median duration of illness — days | 24 | 14 (10, 25) | 26 | 12 (7, 28) |
| Symptoms | | | | |
| Headache | 24 | 24 (100) | 26 | 26 (100) |
| Fever | 24 | 22 (92) | 26 | 23 (88) |
| Neck stiffness | 22 | 20 (91) | 26 | 21 (81) |
| Seizures | 24 | 2 (8) | 26 | 3 (12) |
| Abnormal visual acuity | 22 | 6 (27) | 26 | 4 (15) |
| Papilledema | 21 | 2 (10) | 25 | 1 (4) |
| Glasgow Coma Scale score | 24 | | 26 | |
| 15 | | 19 (79) | | 24 (92) |
| 11–14 | | 5 (21) | | 2 (8) |
| <11 | | 0 (0) | | 0 (0) |
| Cranial nerve palsy | | | | |
| None | 24 | 19 (79) | 26 | 23 (88) |
| Cranial nerve VI | 24 | 4 (17) | 26 | 1 (4) |
| Other cranial nerve | 24 | 1 (4) | 26 | 3 (11) |
| Investigations | | | | |
| Median CSF opening pressure — cm of CSF | 19 | 26.5 (18, 37) | 23 | 24.5 (16, 47) |
| Median CSF white-cell count in HIV infected patients — cells/mm$^3$ | 18 | 38.5 (7, 52) | 20 | 27 (10, 55) |
| Median CSF white-cell count in HIV uninfected patients — cells/mm$^3$ | 5 | 122 (64, 187) | 5 | 94 (45, 117) |
| Median CSF glucose — mmol/l | 24 | 2.47 (1.70, 3.14) | 25 | 2.31 (1.44, 2.76) |
| Median blood glucose — mmol/l | 24 | 5.86 (4.92, 6.84) | 26 | 6.21 (5.11, 7.81) |
| Median CSF: blood glucose ratio | 24 | 0.40 (0.24, 0.53) | 25 | 0.37 (0.16, 0.45) |
| Median CSF fungal count — log10 CFU/ml | 24 | 4.60 (3.90, 5.17) | 26 | 5.16 (3.17, 5.87) |
| Median CD4 count in HIV infected patients — cells/mm$^3$ | 17 | 20 (8, 49) | 21 | 17 (9, 45) |
| Median CD4 count in HIV uninfected patients — cells/mm$^3$ | 5 | 376 (348, 382) | 5 | 504 (305, 968) |
| Median creatinine — mg/dl | 24 | 0.82 (0.66, 1.05) | 26 | 0.78 (0.66, 0.98) |
| QTc interval — ms | 24 | 395.03 (377.55, 410.45) | 26 | 401.20 (374.76, 420.06) |

* Median, interquartile range (IQR) for continuous data and N (%) for categorical data.

shown that tamoxifen had activity in vitro against historical clinical isolates of *C. neoformans*, we found its addition had no impact on EFA. Therefore we do not believe that proceeding to a larger trial, powered to survival, is justified.

It is not clear why tamoxifen did not provide benefit in our patients. The susceptibilities of the *Cryptococcus* isolates from this study to tamoxifen, fluconazole, and amphotericin, were similar to those of isolates from our previous clinical trials (*Hai et al., 2019*; *O'Connor et al., 2020*). However, in contrast with our previous findings we found evidence of synergy when tamoxifen was combined with amphotericin in only 12% (95CI 5%, 24%) of isolates from the trial. This compares with the rate in archived isolates of 67% (95CI 47%, 81%) (*Hai et al., 2019*). Synergy has been suggested as an explanation for the superiority of the amphotericin-flucytosine combination which has delivered improved yeast clearance and survival in a number of trials (*Schwarz et al., 2003*). In this study, we lack sufficient numbers of isolates where tamoxifen-amphotericin synergy is seen to be able to determine whether synergy per se influences EFA.

A second potential explanation is that we may have failed to attain sufficient concentrations of tamoxifen in our patients. We chose a dose of 300 mg/day, based upon the MIC90 of tamoxifen against our historical isolates (16 mg/L) and the expected plasma concentrations this would achieve. Given that tamoxifen is concentrated in the brain (10- to 100-fold), and in macrophage phagosomes, we consider it unlikely that we did not reach drug concentrations greater than the MIC90 at the disease site, although it is possible that absorption of orally administered drug was impaired in our patients.

The rates of adverse events in our study were similar between patients receiving tamoxifen and those in the control arm. Our study was powered to detect a difference in the rate of clearance of

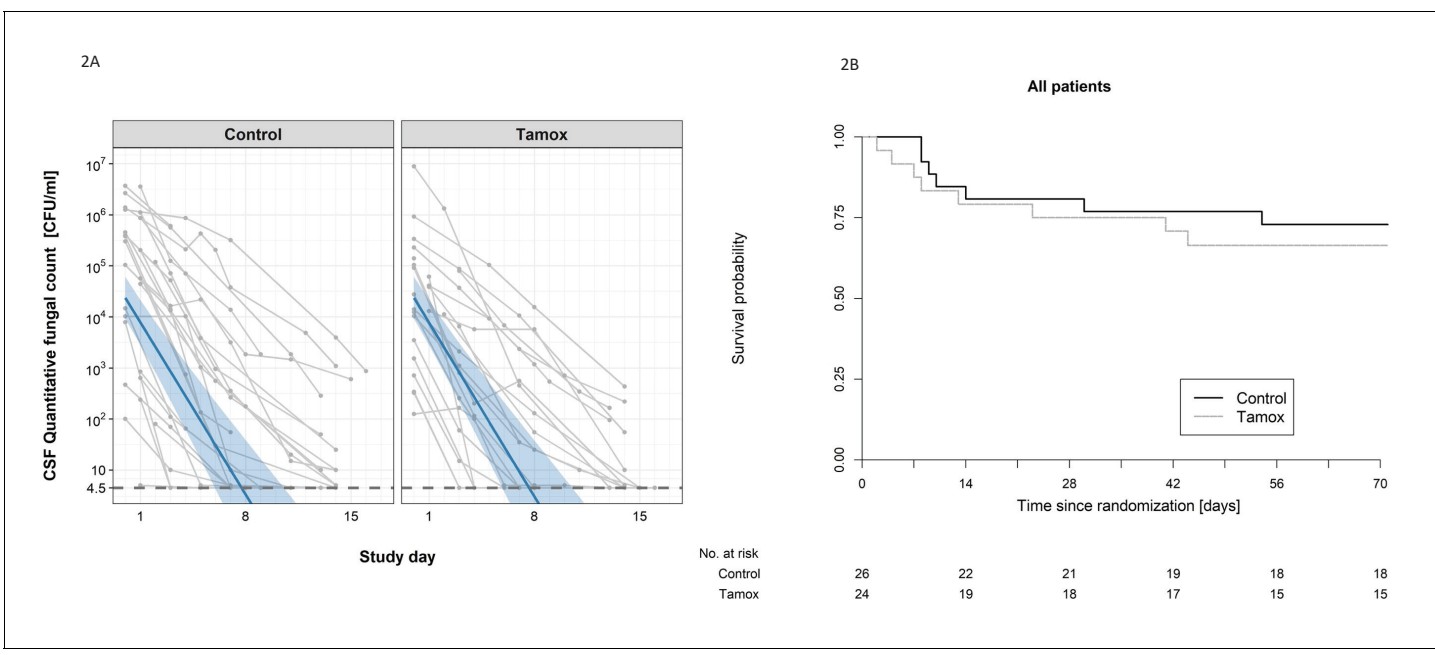

**Figure 2.** The impact of addition of tamoxifen to standard treatment on (**A**) the rate of sterilzation of cerebrospinal fluid, and (**B**) survival until 10 weeks after randomisation. (**A**) Decline in fungal count in CSF as measured in colony-forming units (CFU) per milliliter over the first 2 weeks of treatment by treatment arm. Data from individual patients are shown in grey lines. Bold blue lines show estimated mean with 95% credible intervals (shaded band) of CSF fungal counts based on the joint model described in the statistical analysis. The rate of decline was −0.49 log10CFU/ml/day in patients receiving tamoxifen versus −0.48 log10CFU/ml/day in control patients. The horizontal dashed lines represent the value of detection limit (4.5 CFU/ml). The fitted line crosses the horizontal dashed lines of the detection limit value after day 8 because 25% and 75% of patients had fungal counts under the detection limit at days 8 and 15, respectively. (**B**) Kaplan-Meier survival cures for each study arm over the 10-week study period. Seven death events occurred in the control arm versus 8 in the tamoxifen intervention arm by 10 weeks (estimated risk 27% versus 34%, absolute risk difference = 6.5%) (95% Confidence Interval −19.2 to 32.1%, p = 0.62).

**Table 2.** Primary outcome: Early Fungicidal Activity over the first 2 weeks following randomization (log10 colony-forming units (CFU)/ml/day).

| Analysis populations | Treatment Arm | | Total | | Difference in change | p-value[†] |
|---|---|---|---|---|---|---|
| | Total | Tamoxifen | | Standard of Care | | |
| | N | Change/day (95% CI*) | N | Change/day (95% CI*) | (95% CI*) | |
| Intention-to-treat | 24 | −0.49 (−0.62,−0.37) | 26 | −0.48 (−0.61,−0.37) | −0.005 (−0.16, 0.15) | 0.95 |
| Per-protocol | 23 | −0.48 (−0.61,−0.36) | 25 | −0.48 (−0.61,−0.37) | 0.004 (−0.17, 0.17) | 0.96 |
| HIV-infected patients | 19 | −0.49 (−0.65,−0.37) | 21 | −0.42 (−0.55,−0.31) | −0.072 (−0.25, 0.10) | 0.41 |
| HIV-uninfected patients | 5 | −0.42 (−0.74,−0.21) | 5 | −0.57 (−0.93,−0.33) | 0.16 (−0.18, 0.55) | 0.37 |

*95% CI corresponds to Bayesian 95% credible intervals.

[†]p-value refers to crude 'Wald-type' tests of the mean estimate divided by its standard deviation of the Monte Carlo Markov chain sampling of coefficients derived from the joint model.

yeast from CSF and therefore may have lacked power to detect differences in rates of rarer adverse events. However, there was greater prolongation of the QTc interval in patients on tamoxifen. The mechanism through which tamoxifen causes QT interval prolongation in humans is unknown. In animals there is evidence that the block is multi-channel, due to both inhibition of the $I_{KR}$ and $I_{Ca}$ channels (*Asp et al., 2013*; *He et al., 2003*; *Liu et al., 1998*). Such multi-channel block is considered to confer a reduced risk of life-threatening arrhythmias compared with drugs that block single ion channels. While we did not have any cases of ventricular tachycardia in our study, there was an episode of cardiac arrest in the tamoxifen arm. There are multiple potential causes of cardiac arrest in patients with cryptococcal meningitis, including intracranial pathology and electrolyte disturbances. The cardiac arrest in our study occurred on day 21, 1 week after administration of tamoxifen had finished. However, given tamoxifen's half-life of 5 to 7 days, and the doses used, it is possible that this event was related. Fluconazole is also a recognised cause of QT prolongation. Here, the mechanism

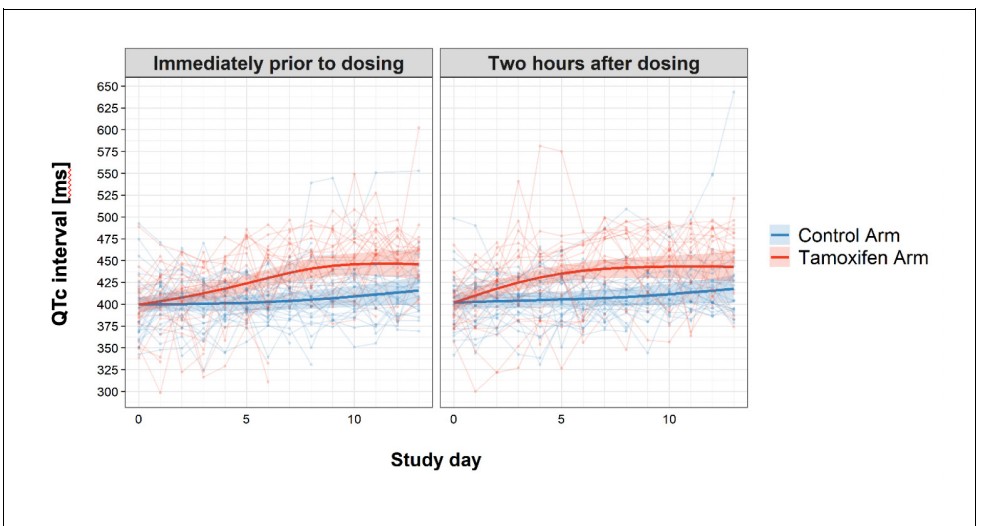

**Figure 3.** Change in QTc interval over the first 2 weeks of treatment by study arm. Faint lines display change in individual patient QTcs; bold lines display the estimated mean and shaded bands the 95% Confidence Intervals; blue = control arm, red = tamoxifen arm. The maximum median difference in the QTc intervals between study arms immediately prior to drug administration was 37.07 ms (95% CI: 21.09, 53.04) and occurred on day 9 of the study. The largest difference in median QTc 2 hr post-drug administration was 33.44 ms (95% CI: 18.67, 48.21) and occurred on day 8 of the study. Additional details regarding change in QTc are provided in the Supplementary Appendix.

**Table 3.** Secondary outcomes: death, disability, and change in CD4 count.

| Death by 10 weeks | Tamoxifen N/total (%) | Control N/total (%) | Risk difference % (95% CI) | p- value* |
|---|---|---|---|---|
| Intention-to-treat population | 8/24 (34) | 7/26 (27) | 6.47 (−19.15, 32.09) | 0.62 |
| Per-protocol population | 7/23 (31) | 6/25 (24) | 6.50 (−18.90, 31.89) | 0.62 |
| HIV infected patients | 7/19 (37) | 6/21 (29) | 8.39 (−20.99, 37.77) | 0.58 |
| HIV uninfected patients | 1/5 (20) | 1/5 (20) | 0.00 (−49.58, 49.58) | 1.00 |
| Disability at 10 weeks | | | | 0.14 |
| Good | 2/23 (9) | 9/25 (36) | | |
| Intermediate | 7/23 (30) | 6/25 (24) | | |
| Severe disability | 6/23 (26) | 3/25 (12) | | |
| Death | 8/23 (35) | 7/25 (28) | | |
| Disability at 10 weeks in HIV infected patients | | | | 0.05 |
| Good | 2/18 (11) | 8/20 (40) | | |
| Intermediate | 5/18 (28) | 6/20 (30) | | |
| Severe disability | 4/18 (22) | 0/20 (0) | | |
| Death | 7/18 (39) | 6/20 (30) | | |
| Disability at 10 weeks in HIV uninfected patients | | | | 0.68 |
| Good | 0/5 (0) | 1/5 (20) | | |
| Intermediate | 2/5 (40) | 0/5 (0) | | |
| Severe disability | 2/5 (40) | 3/5 (60) | | |
| Death | 1/5 (20) | 1/5 (20) | | |
| Change in CD4 count over 10 weeks (cells/uL) | Median Change (IQR) (N) | Median Change (IQR) (N) | | |
| HIV-infected patients | 50.0 (5.00, 142.5) (10) | 40.0 (7.0, 76.0) (13) | | 0.5 |
| HIV-uninfected patients | 393.5 (211.3, 613.8) (4) | −257.5 (−413.7,−171.0) (4) | | 0.02 |

*p-Values not corrected for multiple testing.

is believed to be through modulation of the $I_{kr}$ current of the cardiac depolarization cycle (*Han et al., 2011*). However, we found little evidence of significant QT prolongation in patients in the control arm of our study, and in fact the acute effect of administration of fluconazole was shortening of the QTc interval.

Our experience with tamoxifen is similar to that reported with the anti-depressant drug sertraline. Sertraline has in vitro fungicidal activity against *Cryptococcus neoformans* and a synergistic effect when combined with fluconazole. Results from a pilot dose-finding study of adjunctive sertraline for cryptococcal meningitis suggested it was a safe and potentially effective treatment, although no contemporaneous controls were enrolled in the trial (*Rhein et al., 2016*). Subsequently a large randomized controlled trial powered to mortality was stopped due to futility having enrolled 460 patients (*Rhein et al., 2019*). There was no difference in survival or EFA between the standard therapy or sertraline boosted treatment arms. Of note, a small randomized placebo controled trial from Mexico, published after the phase three trial had begun, found no difference in EFA when sertraline was added to amphotericin and fluconazole, although only 12 patients were enrolled and formal

**Table 4.** Grade 3 or 4 adverse events by 10 weeks.

| Event | Tamoxifen (N = 24) | Control (N = 26) | p-value* |
|---|---|---|---|
| Number of patients with Grade 3 or 4 adverse events (%) | | | |
| Any adverse event | 24 (100) | 26 (100) | 1.0 |
| New neurological events | 9 (38) | 7 (27) | 0.62 |
| New AIDS-defining illness (HIV patients only) | 3 (16) | 5 (24) | 0.58 |
| New cardiac events | 9 (38) | 4 (15) | 0.145 |
| Supraventricular tachycardia | 1 (4) | 0 (0) | 0.48 |
| Ventricular extrasystoles | 3 (13) | 0 (0) | 0.21 |
| Right Bundle Branch Block | 0 (0) | 1 (4) | 1.00 |
| QTc prolongation | 8 (33) | 1 (4) | 0.02 |
| Myocardial infarction | 0 (0) | 1 (4) | 1.00 |
| Cardiac arrest | 1 (4) | 0 (0) | 0.48 |
| Other cardiac adverse events | 1 (4) | 1 (4) | 1.0 |
| Laboratory abnormalities | | | |
| Anemia | 18 (75) | 18 (69) | 0.89 |
| Leukopenia | 2 (8) | 2 (8) | 1.0 |
| Thrombocytopenia | 2 (8) | 4 (15) | 0.74 |
| Elevated aminotransferase | 2 (8) | 4 (15) | 0.74 |
| Raised Creatinine | 3 (13) | 6 (23) | 0.55 |
| Hyperkalemia | 2 (8) | 6 (23) | 0.48 |
| Hypokalemia | 17 (71) | 20 (77) | 0.87 |
| Hyponatremia | 18 (75) | 23 (88) | 0.39 |

*p-Values were not corrected for multiple testing.

statistical testing was not performed (*Villanueva-Lozano et al., 2018*). However, it lends further support for the screening of antifungal treatments in small scale studies using this endpoint.

Other drugs suggested as repurposing candidates for cryptococcal meningitis include the calcium antagonists, such as nifedipine and its sister drugs, used to treat hypertension, and flubendazole, an antihelminthic (*Truong et al., 2018*). Flubendazole is perhaps the most promising of these, appearing to be more potent in vitro than fluconazole, and active against *Cryptococcus* isolates across a range of fluconazole susceptibilities. It crosses the blood brain barrier in mice, but data are lacking regarding humans (*Fok and Kassai, 1998*). While nifedipine crosses the blood brain barrier, it seems unlikley that normal doses and oral administration would reach the plasma levels needed to inhibit *Cryptococcus* growth. However, given our experiences with tamoxifen, and those of others with sertraline, we would caution that better laboratory screening methods than those currently in use are needed to identify potential new treatments for cryptococcal meningitis.

In the mean time, improving access to flucytosine remains a key goal. Progress has been made through efforst to increase generic manufacture through the the Unitaid- Clinton Health Access Initiative for Advanced HIV Disease Initiative's partnership with the Global Fund and the President's Emergency Plan for AIDS Relief. This has resulted in price reductions allowing 2-week treatment courses to be procured for around $100 in some locations.

## Conclusion

Despite apparent in vitro anti-cryptococcal effect including synergy when combined with amphotericin, tamoxifen does not increase the rate of clearance of yeast from cerebrospinal fluid in HIV infected and uninfected patients with cryptococcal meningitis; it is unlikely to result in clinical benefit. Small scale phase two trials such as the one presented here should precede the evaluation of potentially repurposable drugs in clinical endpoint studies. However, the failure of both tamoxifen and sertraline in recent studies underlines the importance of developing novel, specifically anti-

cryptococcal drugs. This will require the support of government and charitable bodies to ensure treatments remain affordable.

## Data access

The original de-identified clinical data underlying the study are available by emailing the OUCRU Data Access Committee at DAC@oucru.org or ekestelyn@oucru.org (Head of the Clinical Trials Unit and Data Access Committee Chair). The review procedures (the data sharing policy and the data request form) are available on the OUCRU website at http://www.oucru.org/data-sharing/.

The statistical code is freely available at https://doi.org/10.5287/bodleian:XmeOzdR8z.

## Acknowledgements

We thank the patients and their relatives for participating in the trial, the members of the data and safety monitoring committee (Professor Tim Peto and Dr Matt Scarborough, University of Oxford) and Dr Nguyen Duc Bang (Pham Ngoc Thach Hospital, Ho Chi Minh City), and the independent trial steering committee (Dr Louise Thwaites and Dr Tan Le Van). This trial was funded through the Wellcome Trust Asia Programme Vietnam Core Grant 106680 and a Wellcome Trust Intermediate Fellowship to JND grant number WT097147MA.

## Additional information

### Funding

| Funder | Grant reference number | Author |
| --- | --- | --- |
| Wellcome Trust | Wellcome Trust Asia Programme Vietnam Core Grant 106680 | Guy E Thwaites |
| Wellcome Trust | Intermediate Fellowship WT097147MA | Jeremy N Day |

The funders had no role in study design, data collection and interpretation, or the decision to submit the work for publication.

### Author contributions

Nguyen Thi Thuy Ngan, Conceptualization, Formal analysis, Investigation, Methodology, Writing - original draft, Project administration; Nhat Thanh Hoang Le, Data curation, Software, Formal analysis, Validation, Visualization, Methodology, Writing - original draft; Nguyen Ngo Vi Vi, Phan Hai Trieu, Nguyen Phu Huong Lan, Nguyen Hoan Phu, William Hope, Investigation, Methodology, Writing - review and editing; Ninh Thi Thanh Van, Validation, Investigation, Project administration; Nguyen Thi Hoang Mai, Data curation, Investigation, Methodology, Writing - review and editing; Duong Van Anh, Conceptualization, Data curation, Investigation, Methodology, Writing - review and editing; Nguyen Van Vinh Chau, Resources, Investigation, Methodology, Writing - review and editing; David G Lalloo, Validation, Writing - review and editing; Justin Beardsley, Luong Thi Hue Tai, Investigation, Writing - review and editing; Nicholas J White, Damian Krysan, Conceptualization, Writing - review and editing; Ronald Geskus, Formal analysis, Writing - review and editing; Guy E Thwaites, Resources, Writing - review and editing; Evelyne Kestelyn, Resources, Data curation, Supervision, Validation, Methodology, Project administration, Writing - review and editing; Tran Quang Binh, Supervision, Methodology, Writing - review and editing; Le Quoc Hung, Supervision, Investigation, Methodology, Writing - review and editing; Nguyen Le Nhu Tung, Resources, Investigation, Methodology, Project administration, Writing - review and editing; Jeremy N Day, Conceptualization, Resources, Data curation, Formal analysis, Supervision, Funding acquisition, Investigation, Methodology, Writing - original draft, Project administration, Writing - review and editing

### Author ORCIDs

Nicholas J White  http://orcid.org/0000-0002-1897-1978
Guy E Thwaites  http://orcid.org/0000-0002-2858-2087

Evelyne Kestelyn [iD] http://orcid.org/0000-0002-5728-0918
Jeremy N Day [iD] https://orcid.org/0000-0002-7843-6280

## Ethics

Clinical trial registration https://clinicaltrials.gov/ct2/show/NCT03112031.

Human subjects: The study protocol was approved by the Ethical Review Committees of the Hospital for Tropical Diseases, Cho Ray Hospital, and the Vietnamese Ministry of Health, and by the Oxford University Tropical Research Ethics Committee. A trial steering committee with 2 independent members oversaw the running of the trial, and an independent data and safety monitoring committee oversaw trial safety. The first safety analysis was performed after the first 20 patients had reached the primary endpoint. The funding bodies and drug manufacturers played no role in the study design, implementation, analysis, or manuscript preparation. All the authors made the decision to submit the manuscript for publication and vouch for the accuracy and completeness of the data and analyses presented. The trial was registered at https://clinicaltrials.gov/ct2/show/NCT03112031.

## Decision letter and Author response

Decision letter https://doi.org/10.7554/eLife.68929.sa1
Author response https://doi.org/10.7554/eLife.68929.sa2

# Additional files

## Supplementary files

• Transparent reporting form

## Data availability

The clinical trial has been conducted in Vietnam under the Ministry of Health and local Ethical Committee approvals. Requests to share the clinical data underlying the trial have to be acknowledged by the local Ethical Committee (and therefore we cannot hand over the data repository or management to an external party). The original de-identified clinical data underlying the study are available by emailing the OUCRU Data Access Committee at DAC@oucru.org or ekestelyn@oucru.org (Head of the Clinical Trials Unit and Data Access Committee Chair). The review procedures (the data sharing policy and the data request form) are available on the OUCRU website at http://www.oucru.org/data-sharing/. The code for the study analysis is freely available at https://doi.org/10.5287/bodleian:XmeOzdR8z.

The following datasets were generated:

| Author(s) | Year | Dataset title | Dataset URL | Database and Identifier |
|---|---|---|---|---|
| Ngan NTT, Thanh HoangLe N, Vi NN, Van NTT, Mai NTH, Anh DV, Trieu PH, Lan NPH, Phu NH, Chau NV, Lalloo DG, Hope W, Beardsley J, White NJ, Geskus R, Thwaites GE, Krysan D, Tai LTH, Kestelyn E, Binh TQ, Hung LQ, Tung NLN, Day JN | 2021 | 28CN: An open label randomized controlled trial of tamoxifen combined with amphotericin B and fluconazole for cryptococcal meningitis | http://www.oucru.org/ | Oxford University Clinical Research Unit, 28CN |
| Ngan NTT, Thanh HoangLe N, Vi NN, Van NTT, Mai NTH, Anh DV, Trieu PH, Lan NPH, Phu NH, Chau NV, Lalloo DG, Hope W, | 2021 | Statistical Analysis Code: An open label randomized controlled trial of tamoxifen combined with amphotericin B and fluconazole for cryptococcal meningitis | https://doi.org/10.5287/bodleian:XmeOzdR8z | Oxford University Research Archive, 10.5287/bodleian:XmeOzdR8z |

Beardsley J, White NJ, Geskus R, Thwaites GE, Krysan D, Tai LTH, Kestelyn E, Binh TQ, Hung LQ, Tung NLN, Day JN

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

# Appendix 1

This appendix has been provided by the authors to give readers additional information about their work.

## Section 1. Statistical analysis plan

## Purpose

This document details the planned analyses and endpoint derivations for the randomized open label trial of tamoxifen combined with amphotericin B and fluconazole for cryptococcal meningitis (NCT03112031) as outlined in the study protocol. It focuses on the analysis for the main clinical trial outcomes and does not include analysis for any subsidiary studies.

## Statistical software

Data derivations will be performed with the statistical software SAS v9.4 (SAS Institute, Cary, North Carolina, US). All statistical analyses will be performed with the statistical software R version R version 3.6.1 (R Foundation for Statistical Computing, Vienna, Austria)[1].

## Interim analyses and early stopping of the trial

Interim analyses for this trial will be conducted by an independent data and safety monitoring board with statistical expertise (Chair: Tim Peto) after the first 20 cases have reached the primary endpoint (completed the first 2 weeks of treatment following randomization or died), as detailed in the study protocol.

Raw data will be transferred from the study statistician (Nhat Le Thanh Hoang) to the Data Safety and Monitoring Board (DSMB) chair and statistician (Tim Peto) in csv format (csv data can be viewed in Excel and imported to any statistical software) together with R code to generate all summary tables.

The trial is not blinded with placebo; however, the clinician investigators of the trial will not be informed of the interim analyses results, but only the decision as to whether to continue the trial or not, and whether any additional safety reporting is needed.

Based on this information, the DSMB chair and statistician will generate the output tables and distribute the interim report amongst the Data Monitoring and Ethics Committee (DMEC) members.

## Analysis populations
### Intention-to treat population (ITT)

The primary analysis population for all analysis is the full analysis population containing all randomized patients except for those mistakenly randomized without cryptococcal meningitis. Patients not receiving any study treatment will still be included in the ITT. Patients will be analyzed according to their randomized arm (intention-to-treat).

### Per-protocol population

The primary endpoint will also be analyzed on the per-protocol population, which will exclude the following patients: major protocol violations and those receiving less than 1 week of administration of the randomized study drug for reasons other than death.

### Derivation rules for the definition of study populations

The following will be considered as 'major protocol violations':

- Pregnancy
- Less than 1 week of amphotericin B antifungal therapy after randomization for reasons other than death (interpreted in the same way as for the study drug, see below). Amphotericin B antifungal therapy is recorded on the antifungal drug (AFDR) form.

- Less than 1 week of administration of tamoxifen study drug for reasons other than death: To allow that study drug is stopped up to 3 days prior to death, this will be interpreted as receiving <7 days of study drug for those who did not die within the first 9 days and as receiving less than [day of death]−3 doses of study drug for those who died earlier (i.e. <six doses for patients who die on day 9, <five doses for patients who die on day 8, ..., no study drug at all for patients who die on days 1–4).

## Baseline characteristics

Baseline characteristics will be summarized as median (interquartile-range (IQR)) for continuous data and n (%) for categorical data. The amount of missing data for each baseline characteristic will also be displayed.

Formal comparisons of baseline characteristics between study arms are discouraged by most statisticians (see e.g. Senn SS (2008): Statistical Issues in Drug Development, 2nd Edition, Wiley [p. 98 f]) but mandated by some journals. To satisfy all potential publishers, we will calculate p-values (based on the Wilcoxon rank sum test for continuous and Fisher's exact test categorical data) but will only report them if mandated by the journal.

Baseline/date of randomization is defined as the date of the first dose of study treatment (AMPHOTERICINBDATESTART where DAY=1 in dataset AFDR). If a subject did not receive any study treatment at all, baseline will be defined as the date of the baseline (history and examination) assessment (BASE.ASSDTC).

The following baseline characteristics will be summarized by treatment arm [with derivation rules in brackets]:

### BASE: baseline – history and examination

All recorded variables in the BASE form with the following modifications:

- - Free text specifications will not be summarized.
- If dates are given (e.g. date of birth, prior HIV diagnosis, or prior to cryptococcal meningitis), the time from that date to baseline will be summarized rather than the date.
- For fluconazole prophylaxis: only yes/no, not the duration will be summarized
- For any antifungal treatment for THIS CURRENT diagnosis of cryptococcal meningitis BEFORE randomization: Only the given antifungals (yes/no), whether it was fluconazole monotherapy (yes/no) and the maximum recorded days on any prior antifungal treatment will be reported.
- Other Opportunistic Infection Prophylaxis up to this admission: Only the given drugs (co-trimoxazole, isoniazid, and/or other) will be summarized.
- Glasgow coma score (GCS) will also be summarized as a categorical variable with values ≤10, 11–14, and 15.
- For visual acuity, the worst result of both eyes will also be summarized.
- Cranial nerve palsies (CNP) will be summarized as 'CNP 6' [CNPLeft6 or CNPRight6 ticked], 'Other CNP' [at least one CNP other than CNP six ticked], 'None' [CNPnone ticked] or 'Unable to assess' [CNPUnableAssess ticked].

### HEMA (laboratory investigations – hematology), CHEMIS (laboratory investigations – biochemistry), MICRO (microbiology), and HIVFU (CD4 and CD8 count)

Baseline results for all values (with proper unit conversion) will be recorded. If no values are available before or at enrolment, values up to one day post enrolment will be used as baseline values for hematology, chemistry, and values up to 14 days post enrolment will be imputed as baseline values for CD4 and CD8. (The latest CD4 value recorded on the Base form will also be included in this derivation as long as it did not occur >3 months [91 days] prior to enrolment.) For chemistry, blood glucose values recorded on the lumbar puncture form will also be included in the derivation.

For microbiology tests, the baseline test result will be summarized as 'positive' if at least one positive test result was recorded up to 3 days post enrolment, and 'negative' if at least one negative and no positive test result was recorded.

### LP (lumbar puncture)

Baseline results for the following values (with proper unit conversion, if necessary) will be recorded: Opening and closing pressure, WCC, % of lymph, % of neut, % of mono, % of eosin, protein, csf glucose, csf/blood glucose ratio, and yeast quantitative count. If no values are available at or before enrolment, values up to 1 day post enrolment will be used as baseline values. For the calculation of the csf/blood glucose ratio, missing blood glucose values on the lumbar puncture form will be imputed with the blood glucose value recorded on the chemistry form if that value is from the same day as the csf glucose value.

Test results for microbiology cerebrospinal fluid (CSF) tests, the baseline test result will be summarized as 'positive' if at least one positive test result was recorded up to 3 days post enrolment, and 'negative' if at least one negative and no positive test result was recorded.

### Imaging (Xray and brainscan)

The number of patients with a chest Xray, a brain magnetic resonance imaging (MRI), or a brain computerized tomography (CT) scan at baseline (allowing −7/+2 days) and the respective numbers of abnormal findings for each imaging method will be summarized.

### Electrocardiogram (ECG) findings

Baseline ECG findings will be presented according to treatment group in terms of heart rate, corrected QT interval (median, IQR, proportion > 500 ms). The QT corrected (QTc) will be classified as 'normal' (<450ms for males, <460ms for females), mildly prolonged ($\geq$450 ms for males or $\geq$460 for females but $\leq$500 ms), and prolonged (>500ms).

Baseline QTc category values will be summarized by treatment arm. Frequency of omitting doses during treatment if QTc remains >500ms will be also summarized by treatment arm.

### Planned analyses

Baseline table for all variables as detailed above for the ITT population will be presented by treatment group.

### Primary endpoint – rate of CSF sterilization during the first 2 weeks

All recorded longitudinal quantitative fungal count measurements up to day 17 (allowing for some delays in the day 14 measurements) will be included in the analysis. Early Fungicidal Activity (EFA) defined as fungal decline (slope) will be modeled based on a linear mixed effects model with longitudinal log10-CSF quantitative culture fungal counts as the outcome, the treatment groups and the time since enrolment, with their interaction, as fixed covariates and random patient-specific intercepts and slopes. The lowest measurable quantitative count is five colony-forming units (CFU)/ml and values below the detection limit (which correspond to recorded values of 0) will be treated as <4.5 CFU/ml, that is non-detectable measurements will be treated as left-censored longitudinal observations. If a patient who misses day 1 measurement completely at random, we will exclude this patient from the analysis; otherwise we keep this patient in the analysis. Based on this model, EFA will be compared between the two treatment arms in all patients (ITT), in the PP population, and subgroups defined by HIV status (uninfected; infected) and baseline fungal burden (<5 log10 CFU/ml; $\geq$ 5 log10 CFU/ml). For the ITT population, the comparison between two arms will also be adjusted for study site and HIV status. Correction for multiple testing (Hochberg procedure as implemented in R function multtest:: mt.rawp2adjp) of all the p-values from the tests for difference effects on EFA between two treatment arms or interaction tests between treatment and subgroups will be provided.

The model will be implemented with the R package Rstan version 2.19.2 which allows to appropriately handle detection limits for longitudinal measurements and also to adjust for the selection bias due to early death in the first 14 days. Stan code will be provided in the appendix. Reported '95% confidence intervals' correspond to Bayesian 95% credible intervals and the reported 'p-values' refer to crude 'Wald-type' tests of the mean estimate divided by its standard deviation. In case Monte Carlo Markov chain diagnostics plots of the fitted stan models indicate

failure of the algorithm we will report results from a mixed model with a detection limit (but ignoring truncation by death) instead and this will be implemented with the R package lmec version 1.0.

## Derivation of overall survival until 10 weeks after randomization

- Definition of time to death: [date of death or censoring] - [date of randomization] + 1
- Definition event indicator: = 1 if patient died = 0 otherwise
- [Date of randomization]: date of the baseline which is derived in Baseline characteristic section
- [Date of death]:
- Final status is death (FINAL.FINALSTT = 2) and the corresponding date of death is FINAL. DEATHDATE2.
- [Date of censoring]:
  - If a final status form is available for the patient (which should be the case for every patient at completion of the study) then the date of censoring is defined as the date of study completion (FINAL.FINALDATE1) or, if the patient did not complete the study, the date of last contact (FINAL.LASTDATE3).
  - If the patient is still under follow-up, that is no final status form is available, the date of censoring is defined as the last recorded date of an inpatient or outpatient assessment, the week 10, a GCS, hematology, or blood chemistry date, or a study drug administration date.

## Planned analysis

- Overall survival will be visualized using Kaplan-Meier curves by treatment arm and displayed with separated panel for each HIV status. The analysis will be based on a Cox proportional hazards regression model with HIV status as stratum variable and treatment arm is the only covariate. We will test for proportional hazards of the treatment effect by means of Schoenfeld residuals (as implemented in R function survival::cox.zph).
- If we have enough event (at least 30 events in total), survival will be modeled with a multivariable Cox regression model including the following covariates in addition to the treatment group: baseline log10-fungal load (modeled linearly), Glasgow coma score less than 15 (yes or no), interaction between HIV infection status and treatment, and Anti Retrovirus (ARV) treatment status at study entry (naive or experienced).

### Subgroup analyses

The following subgroups are pre-defined:

- Per protocol analysis – yes
- HIV serostatus (infected, uninfected)
- Quantitative fungal count at enrolment ($<10^5$ cells/ml, $\geq 10^5$ cells/ml CSF)

Potential heterogeneity of the treatment effect across sub-groups will be tested using likelihood ratio tests for an interaction term between treatment and the grouping variable. All the p-values from these interaction tests and the test of the treatment effect in survival of the ITT population will be corrected for multiple testing (Hochberg procedure as implemented in R function multtest:: mt. rawp2adjp) due to the small sample size of the first interim analysis n=20, we won't perform subgroup analysis and only do it in the final analysis.

### Secondary endpoints-disability at 10 weeks

Derivation: The disability score assessed at week 10 is composed of two sub-scores:

- The 'two simple questions' score [ACTHELP and ISPROBLEM in datasets WEEK10]:
  - If answer to the first question = yes; outcome is classified as 'severe disability'
  - If answer to the second question = yes; outcome is classified as 'intermediate'
  - If answer to both questions = no; outcome is classified as 'good'
- The modified Rankin score: [LB30 in datasets WEEK10]

- ○ If Rankin score = 1; outcome will be classified as 'good'
- ○ If Rankin score = 2 or = 3; outcome will be classified as 'intermediate'
- ○ If Rankin score = 4, = 5 or = 6; outcome will be classified as 'severe disability'

[Note that the Rankin scale is coded as taking values from 1 to 6 on the database, that is, +1 compared to the levels 0–5 according to the published study protocol.]

The worst disability outcome from either questionnaire ('two simple questions' or Rankin score) will be used for analysis. Disability will be defined as 'death' if the patient died before the scheduled time point.

Planned analysis

The ordinal 10-week score ('good'> 'intermediate'> 'severe'> 'death') will be compared between the two arms with a proportional odds logistic regression model depending on the treatment arm and HIV infection status. The result will be summarized as a cumulative odds ratio with corresponding 95% confidence interval and p-value. Patients lost to follow up will be analyzed according to their last recorded disability status. If the fraction of patients lost to follow-up exceeds 10%, we will also perform an alternative analysis based on multiple imputation of missing values. See section Treatment of missing values (multiple imputation).

## Secondary endpoint – clinical adverse events and new laboratory adverse abnormalities

Derivation: Adverse events (AE) are all events recorded on the NEW CARDIAC ADVERSE EVENT (NCAE), NEW NEUROLOGICAL EVENT (NNE), NEW AIDS DEFINING ILLNESS (NADI), or OTHER ADVERSE EVENT (OAE) forms. All grade 3 and 4 AE are collected and will be considered as serious adverse events (SAE); grade 1 and 2 AE are only collected for NCAE, NNE, and NADI events but not OAE.

New laboratory abnormalities are defined as any worsening of a lab value to grade 3 or 4 (including changes from grade 3 to 4) compared to the subject's previous lab value. In addition, to be conservative, if a subject's baseline lab missing value, the worst post-enrolment lab value will be considered a new lab abnormality if it is of grade 3 or 4. A grading table for laboratory abnormalities is provided in the Appendix.

Planned analysis

- Summary of all reported AE – overall (separate summaries by type only and by type and subtype will be produced)
- Summary of all grade 3 and 4 AE – overall and by HIV status
- Summary of grade 3 and 4 AE with onset within the first 2 weeks by type
- Summary of grade 3 and 4 AE with onset during weeks 3–4 by type
- Summary of grade 3 and 4 AE with onset during weeks 5–10 by type
- Summary of total number of grade 3 and 4 AE per patient
- Summary of new laboratory abnormalities

All the summaries will report the frequency of specific adverse events both in terms of the total number of events as well as the number of patients with at least one event. The proportion of patients with at least one such event (overall and for each specific event separately) will be presented and (informally) compared between the two treatment groups based on Fisher's exact test.

## Secondary endpoint - QT prolongation

The QTc will be classified as 'normal' (<450ms for males, <460ms for females), mildly prolonged (≥450 ms for males or ≥460 for females but ≤500 ms) and prolonged (>500ms). All recorded QTc intervals in the first 14 days of treatment and day 21, day 28 following randomization will be included in the analysis. All the measurement after 14 days will be considered as pre-dose measurement and all the QTc measurement values at the time of omitting dose during treatment will be considered as missing values. The main summary measure is the number of patients who has prolonged QTc within the first 14 days and the number of events of QTC prolongation per patient within the first 14 days per arm. The test for the different effect of treatment arm on QTC prolongation will be based on a linear mixed effect model to the QTc data in which will allow for different trends over

the pre-dose and post-dose measurements. The linear mixed effect model was implemented with the R package 'lmec'. In details, we model the relation between time and QTc in a flexible way using restricted cubic splines. We include an interaction term between the treatment arm and both time variables. A random patient-specific intercept and slope is included to account for the heterogeneity of individuals. The model can be written as follows:

$$
\begin{aligned}
Y_i(&t, \text{post}_i, \text{treatment}_i) \\
&= \alpha + a_i + (\text{ns}(t, df = 3) + \text{ns}(t, df = 3) * \text{treatment}_i) * I(\text{post}_i = 0) \\
&\quad + (\text{ns}(t, df = 3) + \text{ns}(t, df = 3) * \text{treatment}_i) * I(\text{post}_i == 1) + b_i * t + \epsilon(t),
\end{aligned}
$$

where,

- $Y_i(t, \text{post}_i, \text{treatment}_i)$: QTc measurement at day $t(t = 0, \ldots, 14,)$, of the pre and 2 hr post-dose measurement in treatment group $\text{treatment}_i$ of patient i,
- $\text{ns}(t, df = 3)$: natural spline function of time with 3 degree of freedom,
- $\text{post} = 0$ if for pre-dose measurement and = 1 for post-dose measurement,
- treatment = 0 for control arm and = 1 for Tamoxifen arm,
- $a_i$ and $b_i$ are random intercept and random slope of the mixed model,
- $\epsilon(t)$ is the measurement error.

Based on this model, longitudinal of QTc measurements will be compared between treatment arms over the first 14 days of treatment following randomization. In addition, we then used the output of the fitted linear mixed effect model to compute the differences in QTc between treatment arm by study day, separately for pre-dose and 2 hr post-dose measurements, based on the delta method[2].

## Analysis of other secondary outcomes

Secondary endpoint – Rate of Immune reconstitution inflammatory syndrome (IRIS) until 10 weeks
   Derivation: The derived endpoint will be the competing risks endpoint of the time to first IRIS or death defined as:
   Time to event = [date of first IRIS event or death or censoring] - [date of randomization] + 1
   Event type:
   0: 'censored': if patient is censored (no IRIS events or death recorded).
   1: 'IRIS': if patient had an IRIS event (any adverse event recorded as IRIS).
   2: 'prior death': if patient died without prior IRIS.
   Planned analysis
   The rate of IRIS will be modeled with cause-specific proportional hazards models with treatment as the only covariate and stratification by HIV infection status, taking into account the competing risk of prior death. Non-parametric estimates of the cumulative incidence functions for the two competing events (IRIS/relapse and prior death) will also be calculated and displayed by treatment arm and tested using Gray log-rank test for sub-distribution hazard.
   Secondary endpoint – Rate of Cryptococcal Meningitis Relapse in the 10 weeks after randomization.
   As for the endpoint 'Rate of IRIS until 10 weeks' (see above) will be analyzed.
   Secondary endpoint – Visual deficit at 10 weeks.
   The visual acuity at 10 weeks is recorded on a six-point scale and will be summarized by treatment arm for each eye separately, and overall where 'overall' is defined as the worst recorded acuity of either eye. The odds of having 'normal acuity' (amongst all surviving patients with a visual assessment) will be informally compared between the treatment arms with a logistic regression model adjusted for HIV status.
   Secondary endpoint – Time to new neurologic event or death until 10 weeks
   Derivation: The derived endpoint will be the competing risks endpoint of the time to first new neurological event or death defined as:
   Time to event = [date of first neurological event or death or censoring] - [date of randomization] + 1
   Event type:
   0/ 'censored': if patient is censored (no neurological event or death recorded).
   1/ 'NNE': if patient had a new neurological event (defined below).
   2/ 'prior death': if patient died without a prior new neurological event.

Neurological events are defined as any grade 3 or 4 new neurological events or any fall in GCS ≥2 points, for ≥ 48 hr (which will also be programmed separately based on recorded longitudinal GCS).

Planned analysis

The time to the first new neurological event or death until 10 weeks will be analyzed in the same way as overall survival.

Secondary endpoint – Longitudinal measurements of intracranial pressure during the first 2 weeks.

This endpoint will be modeled using a mixed effect model as described for the primary outcome.

Secondary endpoint – the change of CD4 cell counts over 10 weeks of survived patients.

The change of CD4 cell counts over 10 weeks of survived patients will be summarized and compared using the Kruskal-Wallis rank sum test, separately for HIV status.

## Other exploratory analyses

Will be performed as appropriate.

## Treatment of missing values (multiple imputation)

Multiple imputation will be performed if do this is the amount of missing values is large. Multiple imputation by chained equations as implemented in the R package 'mice' will be used to deal with missing covariate values for all the Cox regression analysis and the proportional odds logistic regression analysis for disability at 10 weeks. Specifically, 20 imputed sets will be generated and the dataset for multiple imputation will include the following variables:

- Baseline variables: continent, country, age, sex, GCS, on ARV at study entry (no/ yes but ≤ 3 months/ yes, > 3 MONTHS), CD4 cell count
- CSF measurements: opening pressure and yeast quant counts at baseline [both log-transformed]
- Outcomes: overall survival until 10 weeks after randomization, Rankin score at 10 weeks (using method polr in mice package).
- Time-to-event outcomes (i.e. overall survival) will be included as the cumulative (cause-specific) baseline hazard at the observed event or censoring time and an event indicator as recommended by White and Royston (Statist. Med. 2009; 28:1982–1998).

## Additional planned auxiliary analyses

Summary of time to ARV initiation:

- Categorized outcome: On ARVs at study entry/ARVs started after study entry/No ARVs documented.
- Median (IQR) time to ARV initiation in those who started ARV after study entry.
- Details for subjects with no ARVs documented: Subject died within <42 days without ARV/ subject died after >= 42 days without ARV/ subject alive but no ARVs documented.

Number of chest X-rays, CT scans and MRI performed after baseline and proportion with an abnormal result.

Summary whether study drug was terminated before 2 weeks (for reasons other than death) by treatment group – based on tick-box on final status form. Summary of the number of days of tamoxifen treatment after enrolment.

## Grading of laboratory abnormalities

| Laboratory tests | Grade 3 | Grade 4 |
|---|---|---|
| **Hematology** | | |
| Hemoglobin | 6.5–7.9 g/dl | <6.5 g/dl |
| White cell count | 1.0–1.9 K/µl or g/L | <1.0 K/µl or g/L |
| Neutrophils | NEU % xWBC=NEU K/µl: 0.5–1.0 K/µl | NEU % xWBC=NEU K/µl <0.5 K/µl |
| Platelets | 25–50 K/µl or g/L | <25 K/µl or g/L |
| **Biochemistry** | | |
| Sodium - HYPONATRAEMIA | 120–130 mmol/l | <120 mmol/l |
| Sodium - HYPERNATRAEMIA | 155–160 mmol/l | >160 mmol/l |
| Potassium | 2.5–3.0 mmol/l | <2.5 mmol/l |
| Potassium | 6.0–7.0 mmol/l | >7.0 mmol/l |
| Hypocalcemia | 1.5–1.75 mmol/l | <1.5 mmol/l |
| Hypercalcemia | 3.1–3.4 mmol/l | >3.4 mmol/l |
| Hypomagnesemia | 0.3–0.4 mmol/l | <0.3 mmol/l |
| Hypermagnesemia | 1.23–3.3 mmol/l | > 3.3 mmol/l |
| Blood glucose | 1.7–2.2 mmol/l or 30–40 mg/dl<br>13.9–27.8 mmol/l or 250–500 mg/dl | <1.7 mmol/l or < 30 mg/dl<br>>27.8 mmol/l or >500 mg/dl |
| Creatinine | >3X BASELINE OR<br>3–6 X ULN | >6X ULN |
| Aspartate aminotransferase (AST) | >5–20-X ULN | >20X ULN |
| Alanine aminotransferase (ALT) | >5–20-X ULN | >20X ULN |

ULN for Creatinine: 1.36 mg/dL (males), 1.13 mg/dL (females).
ULN for AST/ALT: 40 IU.

## General conventions and mock-up tables/templates
### General conventions

All patients randomized up to the time-point of the interim analysis database snapshot will be included in the tables following an intention-to-treat principle. Tables will contain actual (placebo, tamoxifen) and not masked treatment names.

Statistical tests will report raw p-values and confidence intervals without any adjustment for multiplicity. As described in the protocol stopping for harm of Tamoxifen will be considered if a safety issue emerges which is sufficiently large, in the judgement of the DSMB, to suggest that continued exposure of patients to Tamoxifen is unethical. Early stopping for efficacy of Tamoxifen is not foreseen as this is a pilot study.

As a guidance for stopping early for harm of tamoxifen, the DSMB should consider the following information:

A p-value <0.01 in the direction of harm at an interim analysis.

Clear evidence of harm of tamoxifen in terms of safety or morbidity in the absence of any evidence of a survival benefit due to tamoxifen.

Below are mock-up tables/templates of all information that is planned to be provided to the DMEC.

*Table 1. Summary of patient characteristics at study entry.*

| Characteristic | Tamoxifen (N=XXX) | | Standard treatment (N=XXX) | |
|---|---|---|---|---|
| | N | Summary statistic | N | Summary statistic |
| Age (years) | XXX | XX (XX, XX) | XXX | XX (XX, XX) |
| Sex – male | XX | XX (XX%) | XXX | XX (XX%) |
| Glasgow Coma Score<br>- 15<br>- 11 to 14<br>- 10 or lower | XXX | XX (XX%)<br>XX (XX%)<br>XX (XX%) | XXX | XX (XX%)<br>XX (XX%)<br>XX (XX%) |
| Baseline quantitative fungal count (log10-CFU/ml)<br>QTc (ms)<br>Proportion with QTc>500ms | XXX<br>XX<br>XX | XX (XX, XX)<br>XX<br>XX | XXX<br>XX<br>XX | XX (XX, XX)<br>XX<br>XX |

n refers to the number of patients included in the summary statistic, the summary statistic is the number (%) of patients with the characteristic for categorical data and median (IQR) for continuous data.

**Note:** Table 1 will be generated for all patients, and for patients according to HIV infection status.

*Table 2. Summary of the primary outcome: EFA over the 1 st 2 weeks after randomization.*

| Population | Tamoxifen (N=XXX) | | Placebo (N=XXX) | | Estimated change (95% CI) in log10 CFU/mL of CSF per day |
|---|---|---|---|---|---|
| | N | Summary statistic | N | Summary statistic | |
| All patients (ITT) | XXX | X.XX (X.XX, X.XX) | XXX | X.XX (X.XX, X.XX) | X.XX (X.XX-X.XX); p=X.XX |
| HIV-infected patients | XXX | X.XX (X.XX, X.XX) | XXX | X.XX (X.XX, X.XX) | X.XX (X.XX-X.XX); p=X.XX |
| HIV-uninfected patients | XXX | X.XX (X.XX, X.XX) | XXX | X.XX (X.XX, X.XX) | X.XX (X.XXX.XX); p=X.XX |

*Table 3. Summary of key secondary outcome: survival outcome.*

| Population | No. of deaths | | Hazard ratio (95% CI) | p-value | p-value for heterogeneity* |
|---|---|---|---|---|---|
| | Tamoxifen | Placebo | | | |
| All patients (ITT) | XX/XX | XX/XX | X.XX (X.XX, X.XX) | X.XX | |
| Per protocol populations | XX/XX | XX/XX | X.XX (X.XX, X.XX) | X.XX | |
| HIV status- Infected<br>- Uninfected | XX/XX<br>XX/XX | XX/XX<br>XX/XX | X.XX (X.XX, X.XX)<br>X.XX (X.XX, X.XX) | X.XX<br>X.XX | X.XX |
| Baseline quantitative fungal count<br>- <5 log10 CFU/ml<br>- ≥5 log10 CFU/ml | XX/XX<br>XX/XX | XX/XX<br>XX/XX | X.XX (X.XX, X.XX)<br>X.XX (X.XX, X.XX) | X.XX<br>X.XX | X.XX |

* Hazard ratios and p-values are based on stratified Cox proportional hazards models allowing for separate baseline hazards according to HIV serostatus, except for subgroup analysis of HIV status, model without stratification will be used.

*Figure 1. Kaplan-Meier curves of overall survival by treatment arm.*

Standard Kaplan-Meier curves for the two treatment groups with numbers at risk at the bottom will be displayed. The time axis of the Kaplan-Meier curves will extend to the maximum follow-up duration of 70 days. **Note**: Figure 1 will be generated for all patients only.

*Figure 2. Kaplan-Meier curves of overall survival by treatment arm and stratified HIV status.*

*Table 4. Summary of other secondary outcomes.*

| Outcome | Tamoxifen (N=XXX) | | Placebo (N=XXX) | | Estimate (95% CI); p-value |
|---|---|---|---|---|---|
| | N | Summary statistic | N | Summary statistic | |
| Disability at 10 weeks<br>- Good<br>- Intermediate<br>- Severe disability<br>- Death | XX | XX (XX%)<br>XX (XX%)<br>XX (XX%)<br>XX (XX%) | XX | XX (XX%)<br>XX (XX%)<br>XX (XX%)<br>XX (XX%) | OR of status 'good':<br>X.XX (X.XX-X.XX); p=X.XX |
| Change in QTc | XX | X.XX(X.XX-X.XX) | XX | X.XX(X.XX-X.XX) | Difference in estimated change<br>X.XX (X.XX-X.XX); p=X.XX |
| AUC QTc of the first 2 weeks | | XX(XX%) | | XX(XX%) | Difference in estimated change<br>X.XX (X.XX-X.XX); p=X.XX |
| IRIS | XX | XX (XX%) | XX | XX (XX%) | Cause-specifc HR of IRIS event:<br>X.XX (X.XX-X.XX); p=X.XX |
| Visual deficit at 10 weeks (in survivors)<br>- Normal<br>- Blurred<br>- Finger counting<br>- Movement detection<br>- Light perception<br>- No light perception | XX | XX (XX%)<br>XX (XX%)<br>XX (XX%)<br>XX (XX%)<br>XX (XX%)<br>XX (XX%) | XX | XX (XX%)<br>XX (XX%)<br>XX (XX%)<br>XX (XX%)<br>XX (XX%)<br>XX (XX%) | OR for normal vision:<br>X.XX (X.XX-X.XX); p=X.XX |
| New neurological event or death | XX | XX (XX%) | XX | XX (XX%) | Cause-specifc HR of new neurological event:<br>X.XX (X.XX-X.XX); p=X.XX |
| Relapses | XX | XX (XX%) | XX | XX (XX%) | Cause-specifc HR of relapse event:<br>X.XX (X.XX-X.XX); p=X.XX |
| Intracranial pressure | XX | X.XX(X.XX-X.XX) | XX | X.XX(X.XX-X.XX) | Difference in estimated slope<br>X.XX (X.XX-X.XX); p=X.XX |
| CD4 cell count | XX | X.XX(X.XX-X.XX) | XX | X.XX(X.XX-X.XX) | Difference in estimated change from baseline<br>X.XX (X.XX-X.XX); p=X.XX |

n refers to the number of patients included in the analysis of each outcome. Relapse is defined as need for antifungal treatment intensification or readmission for treatment of cryptococcal disease (as in the protocol). All analyses were done as outlined in the protocol.

**Note:** Table 3 will be generated for all patients and by HIV infection status.

*Table 5. Summary of clinical grade 3 and 4 adverse events.*

| Characteristic | Tamoxifen (N=XX) | | Placebo (N=XX) | | Comparison (p-value) |
|---|---|---|---|---|---|
| | n.pt | n.ae | n.pt | n.ae | |
| Any adverse event | XX (XX%) | XX (XX%) | XX (XX%) | XX (XX%) | X.XX |
| New cardiac event | XX (XX%) | XX (XX%) | XX (XX%) | XX (XX%) | X.XX |
| Neurological event | XX (XX%) | XX (XX%) | XX (XX%) | XX (XX%) | X.XX |
| New AIDS defining illness | XX (XX%) | XX (XX%) | XX (XX%) | XX (XX%) | X.XX |
| … (other collected AE) … | XX (XX%) | XX (XX%) | XX (XX%) | XX (XX%) | X.XX |

n.pt refers to the number of patients with at least one event, n.ae to the total number of adverse event episodes. Comparison between the two groups based on the number of patients with at least one adverse event and Fisher's exact test. **Note:** Table 5 will be generated for all patients. AIDS defining illness will only be defined for patients who are HIV seropositive.

*Table 6. Summary of laboratory grade 3 and 4 adverse events.*
*Table 7. Summary of serious adverse events.*
*Table 8. Summary of unexpected serious adverse events.*
**Note:** Tables 6-8 will have the same layout as Table 5.

## Stan code of the joint model

```
data{
//for longitudinal data
int<lower=1> N_long; // no. rows
int<lower=0,upper=1> y2_censInd[N_long]; // under detection limit index
int<lower=1> I; // no. patient
int<lower=1,upper=I> patid[N_long]; //patid index for random effect
vector [N_long]time_scale; // time_scale censoring data
real log10fc_obs[N_long];// log10fc observing data
real <upper=log10(5)>C;
int<lower=1> p_X;
int<lower=1> p_X_intercept;
int<lower=1> p_X_slope;
vector[p_X]mean_X;
vector<lower=0>[p_X]sd_X;
matrix[N_long,p_X] X_long_scale;//treatment on EFA

// for survival data
matrix[I,p_X_intercept] X_intercept_unscale;//treatment on EFA
matrix[I,p_X_slope] X_slope_unscale;//treatment on EFA;. ~trt.group only
int<lower=0> p_Z;
matrix<lower=0>[I,p_Z] Z; //treatment on survival
real<lower=0,upper=71>ttdeath[I];
int<lower=0> n_time_interval;
vector<lower=0,upper=71>[n_time_interval+1]time_spec;
vector<lower=0,upper=1>[I] death;
matrix<lower=0>[I,n_time_interval]zeros;
int<lower=1,upper=n_time_interval>index_interval[I];
}
parameters{
//for longitudinal data
vector[p_X] fix_eff_scale;//fixed effects for intercept
real<lower=0> sigma; // SD of error measurement
cholesky_factor_corr[2] L_Omega; // prior correlation
vector<lower=0,upper=pi()/2>[2] tau_unif;
//vector<lower=0>[2] tau; // prior scale
matrix[2,I] z;
// for survival data
vector[p_Z] beta;//fixed effects for survival
real<lower=0>lambda[n_time_interval]; // piecewise hazard rate
real eta;// trajectory parameter
real<lower=0> sigma_lambda;
}
transformed parameters {
// for longitudinal data
vector[p_X_slope] b_fix_unscale;//original fixed effects for slope
vector[p_X_intercept] a_fix_unscale;//original fixed effects for intercept
vector[N_long]fit_X_long_scale;
vector[I] a_unscale;// estimated intercept
vector[I] b_unscale;// estimated slope

matrix[I,2] U_raw; // patient random effects
matrix[I,2] U_raw_unscale; // patient random effects
```

```
// for survival data
vector[I] beta_hat;
real test;
vector<lower=0>[2] tau; // prior scale for random effects
for (k in 1:2) tau[k] = 2.5 * tan(tau_unif[k]);

test=sum((fix_eff_scale[2:p_X].*mean_X[2:p_X])./sd_X[2:p_X]);
a_fix_unscale[1]=fix_eff_scale[1]-sum((fix_eff_scale[2:p_X].*mean_X[2:
p_X])./sd_X[2:p_X]);
a_fix_unscale[2:p_X_intercept]=fix_eff_scale[2:p_X_intercept]./sd_X[2:p_X_-
intercept];
b_fix_unscale=(fix_eff_scale[(p_X_intercept+1):p_X])./sd_X[(p_X_intercept
+1):p_X];

U_raw = (diag_pre_multiply(tau,L_Omega) * z)';
for(i in 1:I){
U_raw_unscale[i,1]=U_raw[i,1]-(U_raw[i,2])*mean_X[(p_X_intercept+1)]/sd_X
[(p_X_intercept+1)];
}
U_raw_unscale[,2] = U_raw[,2]/sd_X[(p_X_intercept+1)];
// compute individual intercept
fit_X_long_scale=X_long_scale*fix_eff_scale;
a_unscale=X_intercept_unscale*a_fix_unscale+U_raw_unscale[,1];// contains
only random effects.
// // compute individual slope
for(i in 1:I){
b_unscale[i]=X_slope_unscale[i,]*b_fix_unscale+U_raw_unscale[i,2];
}
//for survival data
beta_hat=Z*beta;
}
model{
vector[N_long] mu;
vector[I] H;//cummulative hazard function;
vector[I] LL;//log density function
matrix[I,n_time_interval] integral_ht;// cummulative hazard function
// Likelihood for longitudinal component
// Set all priors for all parameters of longitudinal component
fix_eff_scale~normal(0,10);
to_vector(z) ~ normal(0,1);
tau_unif ~ uniform(0,pi()/2);
L_Omega ~ lkj_corr_cholesky(2);
for (k in 1:N_long){
mu[k]=fit_X_long_scale[k]+U_raw[patid[k],1]+(U_raw[patid[k],2])*time_scale
[k];
if(y2_censInd[k]==0){
log10fc_obs[k] ~ normal(mu[k],sigma);
}else{
target +=normal_lcdf(C|mu[k],sigma);
}
}
//Likelihood for survival component
//Compute the cumulative hazard function from 0 to ttdeath.
integral_ht=zeros;
```

```
for(i in 1:I){
integral_ht[i,index_interval[i]] = lambda[index_interval[i]]*exp(eta*a_un-
scale[i]+beta_hat[i])*(exp((eta*b_unscale[i])*ttdeath[i])-exp((eta*b_unscale
[i])*time_spec[index_interval[i]]))/(eta*b_unscale[i]);
for(j in 1:(index_interval[i]-1)){
integral_ht[i,j] = lambda[j]*exp(eta*a_unscale[i]+beta_hat[i])*(exp((eta*-
b_unscale[i])*time_spec[j+1])-exp((eta*b_unscale[i])*time_spec[j]))/(eta*-
b_unscale[i]);
}
//Integrated hazard for individual i from 0 to survival time t.surv[i]
H[i]=sum(integral_ht[i,]);// cummulative hazard function
//Survival function
LL[i]=(log(lambda[index_interval[i]])+eta*(a_unscale[i]+b_unscale[i]*ttdeath
[i])+beta_hat[i])*death[i]-H[i];
}
target += sum(LL);
//Set all priors for all parameters of survival component
beta ~ normal(0,10);
if(n_time_interval>2){
lambda[1] ~ lognormal(0,5);
}else{
lambda[1] ~ normal(0,5);
}
for(i in 2:n_time_interval){
lambda[i]~lognormal(lambda[i-1],sigma_lambda);
}
sigma_lambda~cauchy(0,2.5);
eta ~ normal(0,10);
}
generated quantities {
vector[p_X_slope] b_fix_unscale_true;
b_fix_unscale_true[1]=b_fix_unscale[1];
for(i in 2:p_X_slope){
b_fix_unscale_true[i]=b_fix_unscale[1]+b_fix_unscale[i];
```

## Section 2. Data monitoring committee charter
### Data monitoring committee (DMC) overview
### 1. Trial description and study design

Trial number: **28CN**

Trial design: **A randomized trial of Tamoxifen combined with amphotericin B and fluconazole for cryptococcal meningitis**

Trial sponsor: **University of Oxford**

Number of patients: **50**

Names of sites:

| # | Country | City | Name of site | Site number |
|---|---------|------|--------------|-------------|
| 1 | Viet Nam | Ho Chi Minh | Hospital for Tropical Diseases | 03 |
| 2 | Viet Nam | Ho Chi Minh | Cho Ray Hospital | 11 |

Principal Investigators: **Dr Jeremy Day, Dr Nguyen Le Nhu Tung, Dr Le Quoc Hung.**

### 2. DMC terms of reference

This independent DMC has been convened to assess the progress of a clinical study, the safety data and provide recommendations to the sponsor. The members of the DMC serve in an individual

capacity and provide their expertise and recommendations. The DMC will review cumulative study data to evaluate safety, study conduct, and data integrity of the study. This charter will outline the roles and responsibilities and serve as the standard operating procedure (SOP) for the DMC.

1. To consider the data from interim analyses, information from the investigators and relevant information from other sources.
2. In the light of 1, and ensuring that ethical considerations are of prime importance, to report (following each DMC meeting or special meeting if required) to the study sponsor and to recommend on the continuation of the trial.
3. To determine if additional interim analyses of trial data should be undertaken.
4. To consider any requests for release of interim trial data and to recommend on the advisability of this.

## 3. DMC membership

This charter will be agreed by all DMC members Composition of membership will be:

**Chairperson:** Professor Tim Peto (Professor of Medicine, Consultant Physician in Infectious Diseases, General Physician).

**Independent members:** Dr Matt Scarborough (Consultant, Infectious Diseases and General Medicine, OUH NHS trust), Dr Nguyen Duc Bang (Infectious Disease physician, Pham Ngoc Thach Hospital, Ho Chi Minh City, Viet Nam).

## Acronyms

CTU – Clinical Trials Unit (of OUCRU-VN)
DMC – Data Monitoring Committee
OUCRU-VN – Oxford University Clinical Research Unit – Viet Nam
PI – Principal Investigator
TMG – Trial Management Group

## Introduction

The purpose of this charter is to define the roles and responsibilities of the Data Monitoring Committee (DMC), delineate qualifications of the membership, describe the purpose and timing of meetings, provide the procedures for ensuring confidentiality and proper communication, and outline the content of the reports.

The DMC will function in accordance with the ICH guidelines for Good Clinical Practice and the approved trial protocol.

The DMC administration will be coordinated by the OUCRU-VN Clinical Trials Unit. All significant communications, meetings and reports will be made in writing, communicated to all relevant parties and maintained with the Trial Master File.

## Definitions

The following definitions apply to this protocol:
(S)AE

| Table | Definition |
|---|---|
| Adverse Event (AE) | Any untoward medical occurrence in a participant or clinical trial subject to whom an investigational medicinal product has been administered including occurrences that are not necessarily caused by or related to that product. |
| Grade 3 or 4 Adverse Event: | Any untoward medical occurrence of severity defined as grade 3 or 4 by the Common Terminology Criteria for Adverse Events from National Cancer Institute (CTCAE) http://ctep.cancer.gov/protocolDevelopment/ electronic_applications/ctc.htm |

*Continued on next page*

*continued*

| Table | Definition |
|---|---|
| Adverse Reaction (AR) | Any untoward and unintended response to an investigational medicinal product related to any dose administered. |
| Unexpected Adverse Reaction (UAR) | An adverse reaction, the nature or severity of which is not consistent with the information about the investigational |
| | medicinal product in question set out in the Summary of Product Characteristics (SPC) for that product. |
| Serious Adverse Event (SAE) or Serious Adverse Reaction (SAR) or Suspected Unexpected Serious Adverse Reaction (SUSAR) | Respectively any adverse event, adverse reaction or unexpected adverse reaction that: Results in death Is life-threatening* Requires hospitalization or prolongation of existing hospitalization† Results in persistent or significant disability or incapacity Consists of a congenital anomaly or birth defect Is another important medical condition‡ |

*The term life-threatening in the definition of a serious event refers to an event in which the participant is at risk of death at the time of the event; it does not refer to an event that hypothetically might cause death if it were more severe, for example, a silent myocardial infarction.

†Hospitalization is defined as an in-participant admission, regardless of length of stay, even if the hospitalization is a precautionary measure for continued observation. Hospitalizations for a pre-existing condition (including elective procedures that have not worsened) do not constitute an SAE.

‡Medical judgement should be exercised in deciding whether an AE or AR is serious in other situations. The following should also be considered serious: important AEs or ARs that are not immediately life-threatening or do not result in death or hospitalization but may jeopardize the subject or may require intervention to prevent one of the other outcomes listed in the definition above; for example, a secondary malignancy, an allergic bronchospasm requiring intensive emergency treatment, seizures or blood dyscrasias that do not result in hospitalization or development of drug dependency.

**Ethical Committee of Reference:** the lead ethical committee to which all safety reporting and DSMB reports are issued. In the case of this trial, the ethical committee of reference is the Oxford Tropical Research Ethics Committee.

## Roles and responsibilities
### DMC roles and responsibilities

1. This DMC will
   i. Receive, review and feedback when necessary on USAEs reported in detail within 2 weeks of occurrence and followed until resolution
   ii. Meet periodically (see DMC Meetings) to review summary tables of serious adverse events (SAEs), grade 3 and 4 AEs. The DMC may request additional data as required including aggregate and individual subject data related to safety, data integrity and overall conduct of the trial.
   iii. Provide recommendations to continue, modify or terminate the trial depending upon these analyses.
   iv. Communicate other recommendations or concerns as appropriate including requests for additional reviews based on regular reporting and USAE reporting.
   v. Comply with and operate according to the procedures described in this charter.
   vi. Maintain documentation and records of all activities as described below (see DMC Chairman, DMC Meetings, DMC Reports).
   vii. DMC members will have the ability to review unmasked clinical data (this will only be discussed during closed sessions).

 viii. Consider factors external to the study when relevant information becomes available, such as scientific or therapeutic developments that may have an impact on the safety of the participants or the ethics of the study.

 ix. What about review the conduct of the study including protocol violations?

2. DMC Chairman will

 i. Be responsible to archive the interim analysis reports and documentation of rationale for decisions made by the Committee during closed sessions. These will be provided to the Principal Investigator upon completion of the trial.

3. DSMB Statistician will

 i. Generate the analysis tables and distribute the interim report amongst the DSMB members as described below (see section 'Creation of interim analysis reports' below).

## Principal investigator roles and responsibilities

The PI will directly or through delegation:

- Assure the proper conduct of the study including collection of accurate and timely data.
- Compile and report USAEs as described below.
- Promptly report potential safety concern(s) to the DMC.
- Communicate with regulatory authorities, ethical committees and investigators, in a manner that maintains patient safety and integrity of the data.

## DMC participation

Membership will be selected by the Principal Investigator and approved by the trial Sponsor. If a DMC member is unable to continue participation on the board, the reason will be documented and a replacement will be selected by the Principal Investigator with the agreement of the other DMC members and endorsement of the Sponsor.

 DMC members will declare any existing or potential conflicts of interest to the Principal Investigator who will report to the Sponsor. Conflicts of interest will be reduced to the greatest extent that is consistent with assembling an independent and highly competent DMC. Any questions or concerns that arise regarding conflicts of interest will be addressed by the DMC Chair and the Sponsor if necessary. In the case of the Chair having a conflict, by the Sponsor.

 A conflict of interest exists or potentially exists when a member has a personal, professional or financial interest which could unduly influence the member's position with respect to the trial or trial related issues. A conflict of interest should also be addressed if an interest could result in the member's objectivity being questioned by others.

## DMC meetings

### 1. Projected schedule of meetings

Correspondence with the DMC will be initiated by the OUCRU Clinical Trials Unit prior to any subject enrollment in the trial in order for the members to review the charter, to discuss the protocol, agree to the safety reporting procedures, to establish a meeting schedule and to review the study modification and/or termination guidelines. Subsequent interim review meetings will be held to review and discuss interim study data according to the schedule below. Additional meetings may be scheduled at the request of the DMC Chairman or the Sponsor. If scheduled meetings are more than 12 months apart, the DMC Chairman may consider an additional interim review.

| Timeline | Data Review by | Type of Data |
|---|---|---|
| Before study initiation | Entire DMC | Study protocol, safety concerns, DMC Charter and associated procedures/reports |
| After 6 and 12 months of recruitment and yearly thereafter | Entire DMC | Enrolment summary<br>Tables of grade 3 and 4 AEs and SAEs, SARs and SUSARs. Any other requested data |

## 2. Meeting format

DMC meetings will generally be conducted by teleconference and coordinated by the OUCRU-VN CTU. A quorum, defined as a minimum of 2 members (including the Chairman) will be required to hold a DMC meeting. Any member of the DMC may be absent during the meeting provided data tables are circulated in advance and the member has opportunity to forward any concerns to the Chairman before the meeting. Decisions of the DMC should be made by unanimous consensus. However, if this is not possible, majority vote will decide. When appropriate, DMC review sessions may be held by email exchange in lieu of a meeting.

## 3. Open and closed sessions

Sessions may be open (attended by representatives of the sponsor and study team) or closed (attended only by DMC members) at the direction of the DMC. A report based on each DMC meeting will be organized by the Chairman and submitted to the Sponsor. This report will include a recommendation to:

- Continue the trial without modification
- Continue the trial with modification
- Stop the trial due to safety concerns
- Stop the trial for another reason

Reports will be circulated to all DMC members for their approval before being issued.

## 4. Creation and conduct of interim analysis reports

The study statistician will generate the code (in the statistical software R) to generate all tables outlined in the Interim Analysis Plan. The intention is to analyze safety outcomes only to prevent stopping the study when important secondary outcomes including antibiotic use may not yet be clear.

Prior to each interim analysis, raw data will be transferred from the study statistician to the DMC statistician together with R code to generate all summary tables. Based on this information, the DMC statistician generates the tables and distribute the interim report amongst the DMC members.

### Interim analysis plan

All planned analyses will be described in detail in a full Statistical Analysis Plan. This section summarizes the main issues.

## 1. Analysis populations

The primary analysis population for all analyses is the full analysis population containing all randomised patients. Patients will be analysed according to their randomised arm (intention-to-treat). In addition, the primary end point will also be analysed on the per-protocol population, which will exclude the following patients: major protocol violations and those receiving less than 1 week of administration of the randomised study drug for reasons other than death.

## 2. Analysis of primary endpoint

The primary efficacy endpoint is the Early Fungicidal Activity – the rate of decline of yeast counts in cerebrospinal fluid measured over the first 2 weeks of the study. All recorded longitudinal quantitative fungal count measurements up to day 17 (to allow for some delays in the day 14 measurements) will be included in the analysis. EFA will be modeled based on a linear mixed effects model with longitudinal $\log_{10}$-CSF quantitative culture fungal counts as the outcome, interaction terms between the treatment groups and the time since enrolment of the measurement as fixed covariates and random patient-specific intercepts and slopes. The lowest measurable quantitative count is 5 CFU/ml and values below the detection limit (which correspond to recorded values of 0) will be treated as

<5 CFU/ml, i.e. non-detectable measurements will be treated as left-censored longitudinal observations in the analysis as we have done previously. Based on this model, EFA will be compared between the two treatment arms in all patients (intention to treat), in the per-protocol population, and subgroups defined by HIV status (uninfected; infected) and baseline fungal burden (<5 $\log_{10}$ CFU/ml; $\geq 5\log_{10}$ CFU/ml).

## 3. Analysis of secondary efficacy endpoints

*Survival until 10 weeks after randomization*

Overall survival will be visualized using Kaplan-Meier curves and modeled using the Cox proportional hazards regression model with stratification by HIV status. In addition, survival will be modeled with a multivariable Cox regression model including the following covariates in addition to the treatment group: baseline $\log_{10}$-fungal load, Glasgow coma score less than 15 (yes or no), HIV infection status, and ARV treatment status at study entry (naïve or experienced).

*Disability at 10 weeks*

The disability score at week 10 follow-up is defined as the higher (worse) of " the two simple questions" and the Rankin score assessed at that time point, and will be categorized as good outcome, intermediate disability, severe disability, or death (in case the patient died before 10 weeks) as previously described. The ordinal 10-week score ("good"> "intermediate"> "severe"> "death") will be compared between the two arms with a proportional odds logistic regression model depending on the treatment arm and HIV infection status. The result will be summarized as a cumulative odds ratio with corresponding 95% confidence interval and p-value. Patients lost to follow up will be analyzed according to their last recorded disability status. If the rate of patients lost to follow-up exceeds 10%, we will also perform an alternative analysis based on multiple imputation of missing values.

## 4. Analysis of adverse event

The frequency of serious and grade 3&4 adverse reactions as well as the frequency of specific adverse events will be summarized (both in terms of the total number of events as well as the number of patients with at least one event). The proportion of patients with at least one such event (overall and for each specific event separately) will be summarized and (informally) compared between the two treatment groups based on Fisher's exact test.

## 5. Analysis of QTc prolongation

Baseline QTc values will be summarised by patient group. The change in QTc between pre- and 2 hours post-dose of study drug will be summarised, modelled and compared between patient groups over the first 14 days of treatment. QTc intervals at 21 and 28 days following randomisation will be summarised for each treatment group. If outwith the normal range, changes in QTc from baseline will be compared between study groups. The proportion of patients with any episode of QTc>500ms will be compared between study groups.

## 6. Baseline descriptive analyses

Baseline characteristics will be summarized as median (lower and upper quartiles) for continuous data and frequency (percentage) for categorical data. The amount of missing data for each baseline characteristic will also be displayed.

## Study review criteria, stopping rules and guidelines
## 1. Safety analyses

The primary safety endpoint is survival. In addition to the primary safety endpoint, the DMC will consider grade 3 and 4 adverse events, serious adverse events and unexpected or events concerning to the Investigators at the time points defined above.

### 2. Consideration of external data

The DMC will also consider data from other studies or external sources during its deliberations, if available, as these results may have an impact on the status of the patients and design of the current study.

## DMC reports

### 1. Monitoring for safety

The primary charge of the DMC is to monitor patient safety during the study. Formal DMC safety reviews will occur as specified above (see DMC Meetings).

Safety reporting to regulatory and ethical committees will be in accordance with the requirements of each committee and the study protocol.

### 2. Content of DMC reports at formal interim analyses

The detailed content of the interim analysis report will be outlined in a separate document, the Interim Analysis Plan.

### 3. Monitoring for study conduct

The DMC will be updated at each scheduled meeting on study enrolment and major operational issues.

### 4. DMC communication of findings and recommendations

Following each meeting and within 2 weeks of the meeting the chairman will send findings and recommendations of the DMC in writing to the Sponsor. The report should include the date of the meeting, participants, data reviewed by the Committee and a recommendation to continue the trial with/without modification or to stop the trial on a specified basis. The report may include minutes of relevant non- confidential discussion points and any requests for clarification of further information.

These findings and recommendations can result from both the open and closed sessions of the DMC. If these findings include serious and potentially consequential recommendations that require immediate action, the chairman will promptly notify the Principal Investigator and sponsor.

### 5. Response to DMC findings and recommendations

The Sponsor will review and respond to the DMC recommendations. If the DMC recommends continuation of the study without modification, no formal response will be required. If the recommendations request action, such as a recommendation for termination of the study or modification of the protocol, the Sponsor or Principal Investigator will provide a response stating whether the recommendations will be followed and the plan for addressing the issues.

Upon receipt, the DMC will consider the response and will attempt to resolve relevant issues, resulting in a final decision.

The Principal Investigator will disseminate all DMC reports to the relevant ethical committees according to the reporting requirements of that committee.

## DMC study closeout

This study may be terminated based on safety issues or DMC monitoring guidelines. A final study report will be issued to the DMC who may recommend continuing action items to the Sponsor based upon the report.

## Confidentiality

All data provided to the DMC and all deliberations of the DMC will be privileged and confidential. The DMC will agree to use this information to accomplish the responsibilities of the DMC and will not use it for other purposes without written consent from the Sponsor. No communication of the deliberations or recommendations of the DMC, either written or oral, will occur except as required for the DMC to fulfill its responsibilities. Individual DMC members must not have direct communication regarding the study outside the DMC (including, but not limited to the investigators, IRB/EC, regulatory agencies, or sponsor) except as authorized by the DMC.

## Amendments to the DMC charter

This DMC charter can be amended as needed during the course of the study. All amendments will be documented with sequential revision dates, and will be recorded in the report from the DMC meetings. Each revision will be reviewed and agreed upon by the DMC, the Principal Investigator and the Sponsor. All versions of the charter will be archived in the Trial Master File.

## Archiving of DMC activities and related documents

All DMC documentation and records will be retained in the Trial Master File in accordance with local and international regulatory requirements.

## Agreement of DSMB members

Signatures below confirm the agreement of all DSMB members to the contents of this charter and the confidentiality statement above.

Name: Professor Tim Peto Date: Signature:
Name: Dr Matt Scarborough Date: Signature:
Name: Dr Nguyen Duc Bang Date: Signature:

## Agreement of sponsor

Signatures below confirm the agreement of the Sponsor with the contents of this charter.
Name: Evelyne Kestelyn Date: Signature:

## Section 3. The difference in QTc between two study arms over the first 2 weeks of study drug administration

| Study day | Difference of QTC between study arms before drug using (95% CI) | Difference of QTC between study arms 2 hr after drug using (95% CI) |
|---|---|---|
| 0 | 0.00 (0.00, 0.00) | 0.00 (0.00, 0.00) |
| 1 | 3.73 (-0.29, 7.74) | 7.44 (3.45, 11.44) |
| 2 | 7.63 (0.09, 15.18) | 14.51 (6.99, 22.02) |
| 3 | 11.91 (1.75, 22.06) | 20.8 (10.69, 30.92) |
| 4 | 16.73 (5.2, 28.25) | 25.96 (14.47, 37.45) |
| 5 | 22.13 (10.28, 33.97) | 29.68 (17.85, 41.5) |
| 6 | 27.55 (15.32, 39.78) | 32.05 (19.82, 44.27) |
| 7 | 32.29 (18.96, 45.62) | 33.24 (19.92, 46.57) |
| 8 | 35.63 (20.85, 50.42) | 33.44 (18.67, 48.21) |
| 9 | 37.07 (21.09, 53.04) | 32.82 (16.86, 48.77) |

*Continued on next page*

*continued*

| Study day | Difference of QTC between study arms before drug using (95% CI) | Difference of QTC between study arms 2 hr after drug using (95% CI) |
|---|---|---|
| 10 | 36.82 (19.81, 53.83) | 31.54 (14.55, 48.53) |
| 11 | 35.32 (16.92, 53.72) | 29.77 (11.39, 48.15) |
| 12 | 32.97 (12.41, 53.54) | 27.67 (7.13, 48.21) |
| 13 | 30.21 (6.65, 53.77) | 25.41 (1.89, 48.93) |

## Section 4. Adverse events by type and subtype

| Adverse events (AEs) | Tamoxifen (N = 24) | Control (N = 26) | Comparison (p-value) * |
|---|---|---|---|
| Number of patients with adverse events of any grade (%) | | | |
| All AEs combined | 24 (100%) | 26 (100%) | 1 |
| IMMUNE RECONSTITUTION INFLAMMATORY SYNDROME | 0 (0%) | 1 (3.85%) | 1 |
| NEW AIDS DEFINING ILLNESS | 7 (29.17%) | 10 (38.46%) | 0.693 |
| Meningitis tuberculosis | 1 (4.17%) | 1 (3.85%) | 1 |
| Other AIDS events | 1 (4.17%) | 3 (11.54%) | 0.661 |
| Other extrapulmonary tuberculosis | 1 (4.17%) | 0 (0%) | 0.48 |
| Pneumocystis jiroveci pneumonia | 3 (12.5%) | 6 (23.08%) | 0.546 |

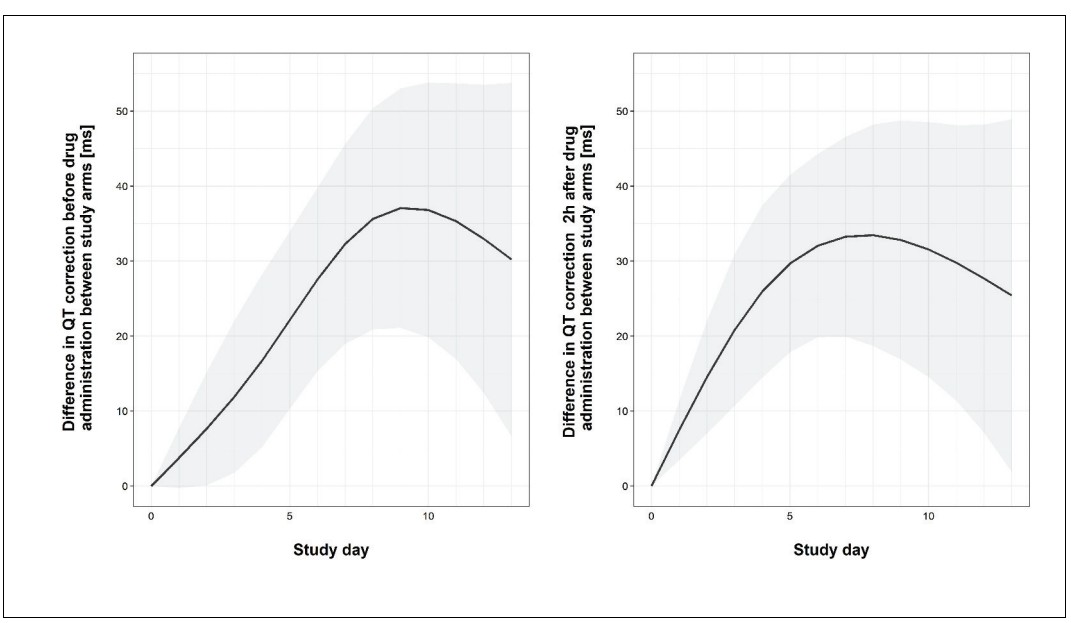

**Appendix 1—figure 1.** Graphical representation of differences in QTc between study arm of ther first 2 weeks of the trial. The bold lines and the shaded bands represent the estimated mean difference with 95% Confidence Interval of QTc between two study arms. The output of the fitted linear mixed effect model computes the differences in QTc between study arms by study day, separately for pre-dose and 2 hours post-dose measurements.

*continued*

| Adverse events (AEs) | Tamoxifen (N = 24) | Control (N = 26) | Comparison (p-value) * |
|---|---|---|---|
| Cerebral toxoplasmosis | 2 (8.33%) | 0 (0%) | 0.225 |
| Pulmonary tuberculosis | 2 (8.33%) | 1 (3.85%) | 0.943 |
| NEW CARDIAC ADVERSE EVENT | 23 (95.83%) | 24 (92.31%) | 1 |
| QRS axis abnormal (New axis deviation) | 3 (12.5%) | 1 (3.85%) | 0.545 |
| Supraventricular tachycardia | 1 (4.17%) | 0 (0%) | 0.48 |
| Ventricular extrasystoles | 8 (33.33%) | 0 (0%) | 0.005 |
| Bundle branch block right | 0 (0%) | 1 (3.85%) | 1 |
| Electrocardiogram QT prolonged | 18 (75%) | 8 (30.77%) | 0.004 |
| Atrioventricular block first degree | 2 (8.33%) | 2 (7.69%) | 1 |
| Myocardial infarction | 0 (0%) | 1 (3.85%) | 1 |
| Sinus tachycardia | 13 (54.17%) | 15 (57.69%) | 1 |
| Cardiac arrest | 1 (4.17%) | 0 (0%) | 0.48 |
| Other cardiac adverse event | 18 (75%) | 13 (50%) | 0.127 |
| Sinus bradycardia | 3 (12.5%) | 3 (11.54%) | 1 |
| NEW NEUROLOGICAL EVENT | 11 (45.83%) | 12 (46.15%) | 1 |
| Brain herniation (coning) | 0 (0%) | 1 (3.85%) | 1 |
| Cranial nerve paralysis | 1 (4.17%) | 1 (3.85%) | 1 |
| Depressed level of consciousness (fall in GCS >= 2 points for >= 48 hr) | 7 (29.17%) | 7 (26.92%) | 1 |
| Headache | 1 (4.17%) | 0 (0%) | 0.48 |
| Hemiplegia/paresis | 1 (4.17%) | 0 (0%) | 0.48 |
| Seizure (fit) | 3 (12.5%) | 5 (19.23%) | 0.793 |
| Other neurological event | 2 (8.33%) | 5 (19.23%) | 0.483 |
| OTHER ADVERSE EVENT | 24 (100%) | 26 (100%) | 1 |
| Hypersensitivity (Allergic reaction) | 3 (12.5%) | 2 (7.69%) | 0.925 |
| Anemia | 18 (75%) | 18 (69.23%) | 0.89 |
| Diarrhea | 3 (12.5%) | 2 (7.69%) | 0.925 |
| Hypertension | 0 (0%) | 2 (7.69%) | 0.491 |
| Hypotension | 2 (8.33%) | 3 (11.54%) | 1 |
| Jaundice | 2 (8.33%) | 0 (0%) | 0.225 |
| Hypokalemia | 17 (70.83%) | 17 (65.38%) | 0.913 |
| Acute Kidney Injury | 0 (0%) | 3 (11.54%) | 0.263 |
| Pleural effusion | 0 (0%) | 1 (3.85%) | 1 |
| Pneumonitis | 5 (20.83%) | 9 (34.62%) | 0.442 |
| Upper gastrointestinal hemorrhage | 0 (0%) | 1 (3.85%) | 1 |
| Vomit | 5 (20.83%) | 3 (11.54%) | 0.61 |
| Other adverse event | 20 (83.33%) | 22 (84.62%) | 1 |

*p-values were not corrected for multiple testing.

## Section 5. Results of drug interactions from two-dimensional chequerboard testing of tamoxifen in combination with either amphotericin, fluconazole

| Antifungal combination | Proportion (%) of isolates where particular drug interactions was observed* | | |
| --- | --- | --- | --- |
| | Synergy FICI ≤ 0.5 | No interaction 0.5 < FICI ≤ 4 | Antagonism FICI > 4 |
| *C. neoformans* | | | |
| Tamoxifen + amphotericin | 11 (5/47) | 89 (42/47) | 0 (0/47) |
| Tamoxifen + fluconazole | 4 (2/47) | 96 (45/47) | 0 (0/47) |
| *C. gattii* | | | |
| Tamoxifen + amphotericin | 33 (1/3) | 67 (2/3) | 0 (0/3) |
| Tamoxifen + fluconazole | 0 (0/3) | 100 (3/3) | 0 (0/3) |

* Numbers in brackets: Numerators are the numbers of strains where interaction was observed; denominators are the numbers of isolates tested.

