## [Decision Letter]

**Acceptance summary:**

This paper will be of particular interest to those in the fields of infectious diseases, HIV, medical mycology, and central nervous system infections; also those interested in drug re-purposing. The main findings, that Tamoxifen, given in addition to the usual antifungal drugs, does not accelerate the rate of clearance of cryptococcal infection from the cerebrospinal fluid in patients with cryptococcal meningitis, and is associated with an increased risk of Cardiac QT prolongation, are well supported by the data. Such small randomised phase II studies are a very useful first clinical step, in order to select only the most promising novel treatments for testing in larger trials.

**Decision letter after peer review:**

Thank you for submitting your article "An open label randomized controlled trial of tamoxifen combined with amphotericin B and fluconazole for cryptococcal meningitis" for consideration by eLife. Your article has been reviewed by 2 peer reviewers, and the evaluation has been overseen by a Reviewing Editor and Jos van der Meer as the Senior Editor. The following individual involved in review of your submission has agreed to reveal their identity: Thomas S. Harrison (Reviewer #1).

Essential revisions:

1) It is slightly confusing as to the different assessments of QT interval. There were no cut offs for grade 3 and 4 QT prolongation as reported in table 4 – could these be added in methods? – a different (I assume) classification of "mildly prolonged" and "prolonged" was found in the statistical analysis plan.

2) In Figure 3 could an average be given for the increase in QT interval when this is maximal? – the reader can read off the graph but I think it would be useful to have the number in manuscript given in the legend.

3) It is surprising that no apparent effect of fluconazole was seen on QT – wasn’t this hard to assess given all participants received the same fluconazole dose?

4) Perhaps in the discussion, it could be mentioned that progress is now being made with flucytosine manufacture, pricing, and access – with generic manufacture now started, and costs reduced to US$70 per 100 x 500 mg tablets.

5) The rate of synergy in the tested isolates was less then in previous reports. Can the authors explain this observation? Is it possible that other cryptococcal genotypes show a higher rate of synergistic interactions? If so should the trial be repeated in areas with a higher rate of synergistic interactions? The previous in vitro interaction study was performed with a similar randomly selected set of Vietnamese isolates so this is perhaps unlikely.

6) Were antagonistic interaction observed? Can the authors present the results of the in vitro interaction study show interactions in a supplemental table.

7) Line 196 " The secondary outcomes are summarized in Table 3". However the secondary: serious adverse effects, were not included in this table.

8) The study was not powered to detect differences in rare serious adverse effects. The absolute number of adverse effects are higher in the tamoxifen arm but most are not significantly different. The authors should discuss this lack of power to detect SE in the discussion.*Reviewer #1:*

The paper describes an open label randomised phase II trial of the treatment of cryptococcal meningitis in Vietnam. 2 weeks of high dose (300mg/day) Tamoxifen was added, or not, to the usual initial antifungal treatment in Vietnam of 2 weeks of amphotericin B deoxycholate plus fluconazole, after which all participants received the usual follow-on treatment with fluconazole alone.

Rate of clearance of infection or early fungicidal activity (EFA) was the same with and without tamoxifen, and tamoxifen was associated with increased QT interval. The authors conclude that there is no justification for a larger study of tamoxifen in this condition.

The study was performed and analysed to a very high standard, and the report is clearly written. The conclusions appear robust – no trend is seen in the EFA, and an effect of tamoxifen on cardiac conduction supported by an increase in the mean QT interval over the first 14 days, as well as by an increased number of participants reaching thresholds levels.

EFA is not a perfect surrogate marker of outcome in cryptococcal meningitis. It cannot capture issues of toxicity (hence the need to assess this separately). And the shape of the association of EFA and outcome may not be linear, such that there may be a threshold above which further increases in EFA do not translate into better outcome. Nevertheless, the authors conclude, and I concur, that such randomised phase II EFA studies are a very useful method for helping select only the most promising novel agents or regimens for testing in larger phase 3 trials.*Reviewer #2:*

The authors describe the results of an open label RCT that evaluates the use of tamoxifen to improve the treatment of cryptococcal meningitis. Using a smart design with a low number of patients, the authors did not find a significant difference in their primary endpoint and conclude that addition of tamoxifen is unlikely to be of clinical benefit.

Overall the manuscript is well written with appropriate use of tables and figures. The introduction is well ordered and gives strong arguments about why tamoxifen is evaluated. The authors describe the beneficial in vitro effects of tamoxifen in combination with other antifungal agents and favorable and clinical achievable pk/pd. Furthermore, the authors give strong arguments for their study design and use of the primary outcome which is defined as Early Fungicidal Activity. The results are clearly presented and the adverse events are well described.

The absolute number of adverse effects are higher in the tamoxifen arm but most are not significantly different. The trial design with a low number of patients is not powered to find differences in adverse events. This study shows that promising preclinical data should first be evaluated using well designed clinical trials.

---

## [Author Response]

Essential revisions:1) It is slightly confusing as to the different assessments of QT interval. There were no cut offs for grade 3 and 4 QT prolongation as reported in table 4 – could these be added in methods? – a different (I assume) classification of "mildly prolonged" and "prolonged" was found in the statistical analysis plan.

Prolonged QTc was classified according to the grading system of the NIH Common

Terminology Criteria for Adverse Events. We have clarified this in the methods lines 125 to 128.

2) In Figure 3 could an average be given for the increase in QT interval when this is maximal? – the reader can read off the graph but I think it would be useful to have the number in manuscript given in the legend.

We have added this information to the legend of Figure 3 (lines 358-362).

3) It is surprising that no apparent effect of fluconazole was seen on QT – wasn’t this hard to assess given all participants received the same fluconazole dose?

The reviewer is correct that all patients, in both treatment arms, received fluconazole. Therefore we can only with rigor ascribe changes in QTc to the use of tamoxifen. We did not detect statistically significant changes in QTc over the first 2 weeks of treatment in patients in the control arm although to the naked eye the curves in figure 3 perhaps seem to rise slightly over the 2 week period. Any change that might occur in QTc seen in the control arm could be due to a number of causes such as accumulating fluconazole, natural disease progression, changes in electrolytes or other treatment (eg amphotericin). However, the notable finding in the control arm is that there was no measurable change in QTc despite the use of fluconazole at robust doses (800mg/day). Of note, the literature describing fluconazole-induced QTc prolongation has been comprised of mostly observational studies and/or case reports. The average time between starting an azole antifungal drug and the development of QTc prolongation is unknown although the appearance of Torsade de Pointes has been reported between 4-6 days (Salem et al., 2017, PMID 29644223). Our data likely reflects the most systematic measurement of QTc with 2 weeks of relatively high dose fluconazole and should provide reassurance about the safety of this drug at these doses.

4) Perhaps in the discussion, it could be mentioned that progress is now being made with flucytosine manufacture, pricing, and access – with generic manufacture now started, and costs reduced to US$70 per 100 x 500 mg tablets.

We have added this information to the discussion (lines 278-282).

5) The rate of synergy in the tested isolates was less then in previous reports. Can the authors explain this observation? Is it possible that other cryptococcal genotypes show a higher rate of synergistic interactions? If so should the trial be repeated in areas with a higher rate of synergistic interactions? The previous in vitro interaction study was performed with a similar randomly selected set of Vietnamese isolates so this is perhaps unlikely.

We have also been puzzled by the lack of synergy seen in isolates from this study. As the reviewer states our previous work included randomly selected clinical isolates, and these included the 3 dominant genotypes causing disease in Vietnam, including VNIa5 (dominant in China), VNIa4 (dominant in western southeast Asia Laos and Thailand) and VNIa93, common in east Africa. We found no association between susceptibility and genotype and therefore don’t think this is likely to explain the findings here. Rather, as we state in lines 276 – 278, we believe we need to develop better tools for determining drug susceptibility in vitro and for screening potential therapeutic compounds.

6) Were antagonistic interaction observed? Can the authors present the results of the in vitro interaction study show interactions in a supplemental table.

We found no evidence of antagonistic interactions observed in susceptibility testing. We have added Section 5 – “Results of drug interactions from two-dimensional

chequerboard testing of tamoxifen in combination with either amphotericin, fluconazole” to the Supplementary materials Page 80.

7) Line 196 " The secondary outcomes are summarized in Table 3". However the secondary: serious adverse effects, were not included in this table.

We have clarified the secondary outcomes reported in Table 3 (death, disabilities, and change in CD4 count) in line 197. Grade 3 and 4 adverse events are reported in Table 4.

8) The study was not powered to detect differences in rare serious adverse effects. The absolute number of adverse effects are higher in the tamoxifen arm but most are not significantly different. The authors should discuss this lack of power to detect SE in the discussion.

We agree with the reviewer – the study was powered to show differences in the rate of sterilisation of cerebrospinal fluid of a magnitude that we believed might be likely to

confer a survival benefit and thus justify a larger trial powered to a survival endpoint. We found no statistically significant difference in number of grade 3 or 4 adverse events between study arms (Table 4) other than prolonged QTc which was consistent with the additional analysis using a linear mixed effect model (Figure 3). However since there is no justification for progression to a larger trial, this is something of a moot point. We have added the sentence ‘Our study was powered to detect a difference in the rate of clearance of yeast from CSF and therefore may have lacked power to detect differences in rates of rarer adverse events’ (lines 246 to 247).